# DSelect-k: Differentiable Selection in the Mixture of Experts with Applications to Multi-Task Learning

**Hussein Hazimeh[1], Zhe Zhao[1], Aakanksha Chowdhery[1], Maheswaran Sathiamoorthy[1]**

**Yihua Chen[1], Rahul Mazumder[2], Lichan Hong[1], Ed H. Chi[1]**

[1]Google, {hazimeh,zhezhao,chowdhery,nlogn,yhchen,lichan,edchi}@google.com
[2]Massachusetts Institute of Technology, rahulmaz@mit.edu

## Abstract

The Mixture-of-Experts (MoE) architecture is showing promising results in improving parameter sharing in multi-task learning (MTL) and in scaling high-capacity neural networks. State-of-the-art MoE models use a trainable "sparse gate" to select a subset of the experts for each input example. While conceptually appealing, existing sparse gates, such as Top-k, are not smooth. The lack of smoothness can lead to convergence and statistical performance issues when training with gradient-based methods. In this paper, we develop DSelect-k: a continuously differentiable and sparse gate for MoE, based on a novel binary encoding formulation. The gate can be trained using first-order methods, such as stochastic gradient descent, and offers explicit control over the number of experts to select. We demonstrate the effectiveness of DSelect-k on both synthetic and real MTL datasets with up to 128 tasks. Our experiments indicate that DSelect-k can achieve statistically significant improvements in prediction and expert selection over popular MoE gates. Notably, on a real-world, large-scale recommender system, DSelect-k achieves over 22% improvement in predictive performance compared to Top-k. We provide an open-source implementation of DSelect-k[1].

## 1 Introduction

The Mixture of Experts (MoE) [14] is the basis of many state-of-the-art deep learning models. For example, MoE-based layers are being used to perform efficient computation in high-capacity neural networks and to improve parameter sharing in multi-task learning (MTL) [32, 22, 21]. In its simplest form, a MoE consists of a set of experts (neural networks) and a trainable gate. The gate assigns weights to the experts on a per-example basis, and the MoE outputs a weighted combination of the experts. This per-example weighting mechanism allows experts to specialize in different partitions of the input space, which has the potential to improve predictive performance and interpretability. In Figure 1 (left), we show an example of a simple MoE architecture that can be used as a standalone learner or as a layer in a neural network.

The literature on the MoE has traditionally focused on softmax-based gates, in which all experts are assigned nonzero weights [17]. To enhance the computational efficiency and interpretability of MoE models, recent works use *sparse gates* that assign nonzero weights to only a small subset of the experts [1, 32, 28, 21]. Existing sparse gates are not differentiable, and reinforcement learning algorithms are commonly used for training [1, 28]. In an exciting work, [32] introduced a new sparse gate (Top-k gate) and proposed training it using stochastic gradient descent (SGD). The ability to train the gate using SGD is appealing because it enables end-to-end training. However, the Top-k gate is not continuous, which can lead to convergence issues in SGD that affect statistical performance (as we demonstrate in our experiments).

---

[1]https://github.com/google-research/google-research/tree/master/dselect_k_moe

35th Conference on Neural Information Processing Systems (NeurIPS 2021).

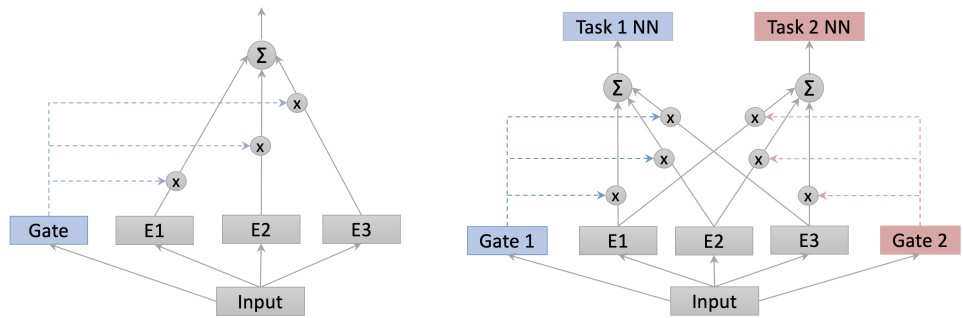

Figure 1: **(Left)**: An example of a MoE that can be used as a standalone learner or layer in a neural network. Here "Ei" denotes the $i$-th expert. **(Right):** A multi-gate MoE for learning two tasks simultaneously. "Task i NN" is a neural network that generates the output of Task i.

In this paper, we introduce DSelect-k: a continuously differentiable and sparse gate for MoE. Given a user-specified parameter $k$, the gate selects at most $k$ out of the $n$ experts. This explicit control over sparsity leads to a cardinality-constrained optimization problem, which is computationally challenging. To circumvent this challenge, we propose a novel, unconstrained reformulation that is equivalent to the original problem. The reformulated problem uses a binary encoding scheme to implicitly enforce the cardinality constraint. We demonstrate that by carefully smoothing the binary encoding variables, the reformulated problem can be effectively optimized using first-order methods such as SGD. DSelect-k has a unique advantage over existing methods in terms of compactness and computational efficiency. The number of parameters used by DSelect-k is logarithmic in the number of experts, as opposed to linear in existing gates such as Top-k. Moreover, DSelect-k's output can be computed efficiently via a simple, closed-form expression. In contrast, state-of-the-art differentiable methods for stochastic k-subset selection and Top-k relaxations, such as [25, 39][2], require solving an optimization subproblem (for each input example) to compute the gate's output.

DSelect-k supports two gating mechanisms: *per-example* and *static*. Per-example gating is the classical gating technique used in MoE models, in which the weights assigned to the experts are a function of the input example [14, 32]. In static gating, a subset of experts is selected and the corresponding weights do not depend on the input [28]. Based on our experiments, each gating mechanism can outperform the other in certain settings. Thus, we study both mechanisms and advocate for experimenting with each.

MTL is an important area where MoE models in general, and our gate in particular, can be useful. The goal of MTL is to learn multiple tasks simultaneously by using a shared model. Compared to the usual single task learning, MTL can achieve better generalization performance through exploiting task relationships [4]. One key problem in MTL is how to share model parameters between tasks [30]. For instance, sharing parameters between unrelated tasks can potentially degrade performance. The multi-gate MoE [22] is a flexible architecture that allows for learning what to share between tasks. Figure 1 (right) shows an example of a multi-gate MoE, in the simple case of two tasks. Here, each task has its own gate that adaptively controls the extent of parameter sharing. In our experiments, we study the effectiveness of DSelect-k in the context of the multi-gate MoE.

**Contributions:** On a high-level, our main contribution is DSelect-k: a new continuously differentiable and sparse gate for MoE, which can be directly trained using first-order methods. Our technical contributions can be summarized as follows. **(i)** The gate selects (at most) $k$ out of the $n$ experts, where $k$ is a user-specified parameter. This leads to a challenging, cardinality-constrained optimization problem. To deal with this challenge, we develop a novel, unconstrained reformulation, and we prove that it is equivalent to the original problem. The reformulation uses a binary encoding scheme that implicitly imposes the cardinality constraint using learnable binary codes. **(ii)** To make the unconstrained reformulation smooth, we relax and smooth the binary variables. We demonstrate that, with careful initialization and regularization, the resulting problem can be optimized with first-order methods such as SGD. **(iii)** We carry out a series of experiments on synthetic and real MTL datasets, which show that our gate is competitive with state-of-the-art gates in terms of parameter sharing and predictive performance. **(iv)** We provide an open-source implementation of DSelect-k.

---

[2]These methods were not designed specifically for the MoE.

## 1.1 Related Work

**MoE and Conditional Computation:** Since MoE was introduced by [14], an exciting body of work has extended and studied this model, e.g., see [17, 13, 16]. Recently, MoE-based models are showing success in deep learning. For example, [32] introduced the sparse Top-k gate for MoE and showed significant computational improvements on machine translation tasks; we discuss exact connections to this gate in Section 2. The Top-k gate has also been utilized in several state-of-the-art deep learning models that considered MTL tasks, e.g., [21, 27, 9]. Our work is also related to the conditional computation models that activate parts of the neural network based on the input [2, 1, 32, 12, 35]. Unlike DSelect-k, these works are based on non-differentiable models, or heuristics where the training and inference models are different.

**Stochastic k-Subset Selection and Top-k Relaxation**: A related line of work focuses on stochastic k-subset selection in neural networks, e.g., see [25, 5, 38] and the references therein. Specifically, these works propose differentiable methods for sampling $k$-subsets from a categorical distribution, based on extensions or generalizations of the Gumbel-softmax trick [23, 15]. However, in the MoE we consider *deterministic* subset selection—determinism is a common assumption in MoE models that can improve interpretability and allows for efficient implementations [14, 17, 32]. In contrast, the stochastic approaches described above are suitable in applications where there is an underlying sampling distribution, such as in variational inference [19]. Another related work is the differentiable relaxation of the Top-k operator proposed by [39]. All the aforementioned works perform dense training (i.e., the gradients of all experts, even if not selected, will have to be computed during backpropagation), whereas DSelect-k can (to an extent) exploit sparsity to speed up training, as we will discuss in Sections 2 and 3. Moreover, the stochastic k-subset selection framework in [25] (which encompasses several previous works) and the Top-k relaxation in [39] require solving an optimization subproblem to compute the gate output–each example will require solving a separate subproblem in the per-example gating setting, which can be computationally prohibitive. In contrast, DSelect-k's output is computed efficiently via a closed-form expression.

**Sparse Transformations to the Simplex:** These are sparse variants of the softmax function that can output sparse probability vectors, e.g., see [24, 26, 6, 3]. While similar to our work in that they output sparse probability vectors, these transformations cannot control the sparsity level precisely as DSelect-k does (through a cardinality constraint). Thus, these transformations may assign some examples or tasks sparse combinations and others dense combinations.

**MTL:** In Appendix A, we review related literature on MTL.

## 2 Gating in the Mixture of Experts

In this section, we first review the MoE architecture and popular gates, and then discuss how these gates compare to our proposal. We will assume that the inputs to the MoE belong to a space $\mathcal{X} \subset \mathbb{R}^p$. In its simplest form, the MoE consists of a set of $n$ experts (neural networks) $f_i : \mathcal{X} \to \mathbb{R}^u$, $i \in \{1, 2, \ldots, n\}$, and a gate $g : \mathcal{X} \to \mathbb{R}^n$ that assigns weights to the experts. The gate's output is assumed to be a probability vector, i.e., $g(x) \geq 0$ and $\sum_{i=1}^{n} g(x)_i = 1$, for any $x \in \mathcal{X}$. Given an example $x \in \mathcal{X}$, the corresponding output of the MoE is a weighted combination of the experts:

$$\sum_{i=1}^{n} f_i(x) g(x)_i. \tag{1}$$

Next, we discuss two popular choices for the gate $g(.)$ that can be directly optimized using SGD.

**Softmax Gate:** A classical model for $g(x)$ is the softmax gate: $\sigma(Ax + b)$, where $\sigma(.)$ is the softmax function, $A \in \mathbb{R}^{n \times p}$ is a trainable weight matrix, and $b \in \mathbb{R}^n$ is a bias vector [17]. This gate is dense, in the sense that all experts are assigned nonzero probabilities. Note that static gating (i.e., gating which does not depend on the input example) can be obtained by setting $A = 0$.

**Top-k Gate:** This is a sparse variant of the softmax gate that returns a probability vector with only k nonzero entries [32]. The Top-k gate is defined by $\sigma(KeepTopK(Ax + b))$, where for any vector $v$, $KeepTopK(v)_i := v_i$ if $v_i$ is in the top k elements of $v$, and $KeepTopK(v)_i := -\infty$ otherwise[3]. This gate is conceptually appealing since it allows for direct control over the number of experts to select

---

[3]To balance the load across experts, [32] add noise and additional regularizers to the model.

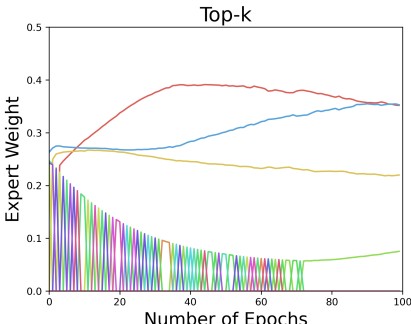
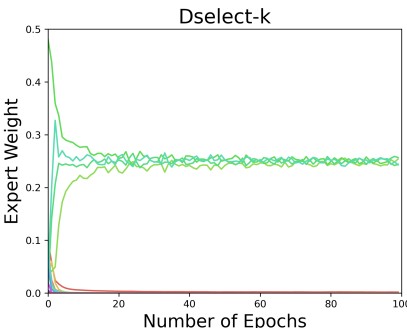

Figure 2: Expert weights output by Top-k (left) and DSelect-k (right) during training on synthetic data generated from a MoE, under static gating. Each color represents a separate expert. Here DSelect-k recovers the true experts used by the data-generating model, whereas Top-k does not recover and exhibits oscillatory behavior. See Appendix C.2 for details on the data and setup.

and is trained using SGD. Moreover, the Top-k gate supports *conditional training*: in backpropagation, for each input example, only the gradients of the loss w.r.t. top k elements need to be computed. With a careful implementation, conditional training can lead to computational savings. However, the Top-k gate is not continuous, which implies that the gradient does not exist at certain inputs. This can be problematic when training is done using gradient-based methods. To gain more insight, in Figure 2 (left), we plot the expert weights chosen by the Top-k gate during training with SGD. The results indicate an oscillatory behavior in the output of the Top-k gate, which can be attributed to its discontinuous nature: a small change in the input can lead to "jumps" in the output.

**Comparison with DSelect-k:** We develop DSelect-k in Section 3. Here we present a high-level comparison between DSelect-k and Top-k. Similar to Top-k, DSelect-k can select $k$ out of the $n$ experts and can be trained using gradient-based optimization methods. A major advantage of DSelect-k over Top-k is that it is continuously differentiable, which leads to more stable selection of experts during training—see Figure 2 (right). During inference, DSelect-k only needs to evaluate a subset of the experts, which can lead to computational savings. However, DSelect-k supports conditional training only partially. At the start of training, it uses all the available experts, so conditional training is not possible. As we discuss in Section 3, after a certain point during training, DSelect-k converges to a small subset of the experts, and then conditional training becomes possible. Our experiments indicate that DSelect-k can have a significant edge over Top-k in terms of prediction and expert selection performance, so the full support for conditional training in Top-k seems to come at the expense of statistical performance.

## 3 Differentiable and Sparse Gating

In this section, we develop DSelect-k, for both the static and per-example gating settings. First, we introduce the problem setup and notation. To simplify the presentation, we will develop the gate for a single supervised learning task, and we note that the same gate can be used in MTL models. We assume that the task has an input space $\mathcal{X} \subset \mathbb{R}^p$, an output space $\mathcal{Y}$, and an associated loss function $\ell : \mathcal{Y} \times \mathbb{R} \to \mathbb{R}$. We denote the set of $N$ training examples by $\mathcal{D} = \{(x_i, y_i) \in \mathcal{X} \times \mathcal{Y}\}_{i=1}^N$. We consider a learning model defined by the MoE in Equation (1). For simplicity, we assume that the experts are scalar-valued and belong to a class of continuous functions $\mathcal{H}$. We assume that the number of experts $n = 2^m$ for some integer $m$—in Appendix B.2, we discuss how the gate can be extended to arbitrary $n$. For convenience, given a non-negative integer $i$, we denote the set $\{1, 2, \ldots, i\}$ by $[i]$.

In Section 3.1, we develop DSelect-k for static gating setting. Then, in Section 3.2, we generalize it to the per-example setting.

### 3.1 DSelect-k for Static Gating

Our goal here is to develop a static gate that selects a convex combination of at most $k$ out of the $n$ experts. The output of the gate can be thought of as a probability vector $w$ with at most $k$ nonzero entries, where $w_i$ is the weight assigned to the expert $f_i$. A natural way to minimize the empirical risk of the MoE model is by solving the following problem:

$$\min_{f_1,\dots f_n,w} \quad \frac{1}{N}\sum_{(x,y)\in\mathcal{D}}\ell\Big(y,\sum_{i=1}^{n}f_i(x)w_i\Big) \tag{2a}$$

$$\text{s.t.} \quad \|w\|_0 \leq k \tag{2b}$$

$$\sum_{i=1}^{n}w_i = 1, \ w \geq 0. \tag{2c}$$

In the above, the $L_0$ norm of $w$, $\|w\|_0$, is equal to the number of nonzero entries in $w$. Thus, the cardinality constraint (2b) ensures that the gate selects at most $k$ experts. Problem (2) is a combinatorial optimization problem that is not amenable to SGD due to the cardinality constraint (2b) and the simplex constraints in (2c). In what follows of this section, we first transform Problem (2) into an equivalent unconstrained optimization problem, based on a binary encoding scheme. However, the unconstrained problem cannot be directly handled using SGD due to the presence of binary variables. Thus, in a second transformation, we smooth the binary variables, which leads to an optimization problem that is amenable to SGD.

**Road map:** In Section 3.1.1, we introduce the *single expert selector*: a construct for choosing 1 out of $n$ experts by using binary encoding. In Section 3.1.2, we leverage the single expert selector to transform Problem (2) into an unconstrained one. Then, in Section 3.1.3, we smooth the unconstrained problem and discuss how SGD can be applied.

### 3.1.1   Single Expert Selection using Binary Encoding

The single expert selector (selector, for short) is a fundamental construct that we will later use to convert Problem (2) to an unconstrained optimization problem. At a high-level, the single expert selector chooses the index of 1 out of the $n$ experts and returns a one-hot encoding of the choice. For example, in the case of 4 experts, the selector can choose the first expert by returning the binary vector $[1\ 0\ 0\ 0]^T$. Generally, the selector can choose any of the experts, and its choice is determined by a set of binary encoding variables, as we will describe next.

The selector is parameterized by $m$ (recall that $m = \log_2 n$) binary variables, $z_1, z_2, \dots, z_m$, where we view these variables collectively as a binary number: $z_m z_{m-1}\dots z_1$. The integer represented by the latter binary number determines which expert to select. More formally, let $l$ be the integer represented by the binary number $z_m z_{m-1}\dots z_1$. The selector is a function $r : \mathbb{R}^m \to \{0,1\}^n$ which maps $z := [z_1, z_2, \dots, z_m]^T$ to a one-hot encoding of the integer $(l+1)$. For example, if all the $z_i$'s are 0, then the selector returns a one-hot encoding of the integer 1. Next, we define the selector $r(z)$. For easier exposition, we start with the special case of 4 experts and then generalize to $n$ experts.

**Special case of** 4 **experts:** In this case, the selector uses two binary variables $z_1$ and $z_2$. Let $l$ be the integer represented by the binary number $z_2 z_1$. Then, the selector should return a one-hot encoding of the integer $(l+1)$. To achieve this, we define the selector $r(z)$ as follows:

$$r(z) = [\bar{z}_1\bar{z}_2, \ z_1\bar{z}_2, \ \bar{z}_1 z_2, \ z_1 z_2]^T \tag{3}$$

where $\bar{z}_i := 1 - z_i$. By construction, exactly one entry in $r(z)$ is 1 (specifically, $r(z)_{l+1} = 1$) and the rest of the entries are zero. For example, if $z_1 = z_2 = 0$, then $r(z)_1 = 1$ and $r(z)_i = 0, i \in \{2, 3, 4\}$.

**General case of** $n$ **experts:** Here we generalize the selector $r(z)$ to the case of $n$ experts. To aid in the presentation, we make the following definition. For any non-negative integer $l$, we define $\mathcal{B}(l)$ as the set of indices of the nonzero entries in the binary representation of $l$ (where we assume that the least significant bit is indexed by 1). For example, $\mathcal{B}(0) = \emptyset$, $\mathcal{B}(1) = \{1\}$, $\mathcal{B}(2) = \{2\}$, and $\mathcal{B}(3) = \{1, 2\}$. For every $i \in [n]$, we define the $i$-th entry of $r(z)$ as follows:

$$r(z)_i = \prod_{j\in\mathcal{B}(i-1)} (z_j) \prod_{j\in[m]\setminus\mathcal{B}(i-1)} (1 - z_j) \tag{4}$$

In the above, $r(z)_i$ is a product of $m$ binary variables, which is equal to 1 iff the integer $(i-1)$ is represented by the binary number $z_m z_{m-1}\dots z_1$. Therefore, $r(z)$ returns a one-hot encoding of the index of the selected expert. Note that when $n = 4$, definitions (3) and (4) are equivalent.

### 3.1.2   Multiple Expert Selection via Unconstrained Minimization

In this section, we develop a combinatorial gate that allows for transforming Problem (2) into an unconstrained optimization problem. We design this gate by creating $k$ instances of the single expert

selector $r(.)$, and then taking a convex combination of these $k$ instances. More formally, for every $i \in [k]$, let $z^{(i)} \in \{0, 1\}^m$ be a (learnable) binary vector, so that the output of the $i$-th instance of the selector is $r(z^{(i)})$. Let $Z$ be a $k \times m$ matrix whose $i$-th row is $z^{(i)}$. Moreover, let $\alpha \in \mathbb{R}^k$ be a vector of learnable parameters. We define the *combinatorial gate q* as follows:

$$q(\alpha, Z) = \sum_{i=1}^{k} \sigma(\alpha)_i r(z^{(i)}),$$

where we recall that $\sigma(.)$ is the softmax function. Since for every $i \in [k]$, $r(z^{(i)})$ is a one-hot vector, we have $\|q(\alpha, Z)\|_0 \leq k$. Moreover, since the weights of the selectors are obtained using a softmax, we have $q(\alpha, Z) \geq 0$ and $\sum_{i=1}^{n} q(\alpha, Z)_i = 1$. Thus, $q(\alpha, Z)$ has the same interpretation of $w$ in Problem (2), without requiring any constraints. Therefore, we propose replacing $w$ in the objective of Problem (2) with $q(\alpha, Z)$ and removing all the constraints. This replacement leads to an equivalent unconstrained optimization problem, as we state in the next proposition.

**Proposition 1.** *Problem* (2) *is equivalent*[4] *to:*

$$\min_{f_1, \dots f_n, \alpha, Z} \quad \frac{1}{N} \sum_{(x,y) \in \mathcal{D}} \ell\Big(y, \sum_{i=1}^{n} f_i(x) q(\alpha, Z)_i\Big) \tag{5}$$
$$z^{(i)} \in \{0, 1\}^m, \ i \in [k]$$

The proof of Proposition 1 is in Appendix B.1. Unlike Problem (2), Problem (5) does not involve any constraints, aside from requiring binary variables. However, these binary variables cannot be directly handled using first-order methods. Next, we discuss how to smooth the binary variables in order to obtain a continuous relaxation of Problem (5).

### 3.1.3 Smooth Gating

In this section, we present a procedure to smooth the binary variables in Problem (5) and discuss how the resulting problem can be optimized using first-order methods. The procedure relies on the *smooth-step* function, which we define next.

**Smooth-step Function:** This is a continuously differentiable and S-shaped function, similar in shape to the logistic function. However, unlike the logistic function, the smooth-step function can output 0 and 1 exactly for sufficiently large magnitudes of the input. The smooth-step and logistic functions are depicted in Appendix B.3. More formally, given a non-negative scaling parameter $\gamma$, the smooth-step function, $S : \mathbb{R} \to \mathbb{R}$, is a cubic piecewise polynomial defined as follows:

$$S(t) = \begin{cases} 0 & \text{if } t \leq -\gamma/2 \\ -\frac{2}{\gamma^3} t^3 + \frac{3}{2\gamma} t + \frac{1}{2} & \text{if } -\gamma/2 \leq t \leq \gamma/2 \\ 1 & \text{if } t \geq \gamma/2 \end{cases}$$

The parameter $\gamma$ controls the width of the fractional region (i.e., the region where the function is strictly between 0 and 1). Note that $S(t)$ is continuously differentiable at all points—this follows since at the boundary points $\pm\gamma/2$, we have: $S'(-\gamma/2) = S'(\gamma/2) = 0$. This function has been recently used for conditional computation in soft trees [11] and is popular in the computer graphics literature [8, 29].

**Smoothing:** We obtain DSelect-k from the combinatorial gate $q(\alpha, Z)$ by (i) relaxing every binary variable in $Z$ to be continuous in the range $(-\infty, +\infty)$, i.e., $Z \in \mathbb{R}^{k \times m}$, and (ii) applying the smooth-step function to $Z$ element-wise. Formally, DSelect-k is a function $\tilde{q}$ defined as follows:

$$\tilde{q}(\alpha, Z) := q(\alpha, S(Z)) = \sum_{i=1}^{k} \sigma(\alpha)_i r\big(S(z^{(i)})\big), \tag{6}$$

where the matrix $S(Z)$ is obtained by applying $S(\cdot)$ to $Z$ element-wise. Note that $\tilde{q}(\alpha, Z)$ is continuously differentiable so it is amenable to first-order methods. If $S(Z)$ is binary, then $\tilde{q}(\alpha, Z)$

---

[4]Equivalent means that the two problems have the same optimal objective, and given an optimal solution for one problem, we can construct an optimal solution for the other.

selects at most $k$ experts (this holds since $\tilde{q}(\alpha, Z) = q(\alpha, S(Z))$, and from Section 3.1.2, $q$ selects at most $k$ experts when its encoding matrix is binary). However, when $S(Z)$ has any non-binary entries, then more than $k$ experts can be potentially selected, meaning that the cardinality constraint will not be respected. In what follows, we discuss how the gate can be optimized using first-order methods, while ensuring that $S(Z)$ converges to a binary matrix so that the cardinality constraint is enforced.

We propose using $\tilde{q}(\alpha, Z)$ in MoE, which leads to the following optimization problem:

$$\min_{f_1,\dots f_n,\alpha,Z} \quad \frac{1}{N}\sum_{(x,y)\in\mathcal{D}} \ell\Big(y, \sum_{i=1}^{n} f_i(x)\tilde{q}(\alpha, Z)_i\Big). \tag{7}$$

Problem (7) can be viewed as a continuous relaxation of Problem (5). If the experts are differentiable, then the objective of Problem (7) is differentiable. Thus, we propose optimizing MoE end-to-end using first-order methods. We note that $\tilde{q}(\alpha, Z)$ uses $(k + k \log n)$ learnable parameters. In contrast, the Top-k and softmax gates (discussed in Section 2) use $n$ parameters. Thus, for relatively small $k$, our proposal uses a smaller number of parameters. Next, we discuss how DSelect-k's parameters should be initialized in order to ensure that it is trainable.

**Initialization:** By the definition of the smooth-step function, if $S(Z_{ij})$ is binary then $S'(Z_{ij}) = 0$, and consequently $\frac{\partial \ell}{\partial Z_{ij}} = 0$. This implies that, during optimization, if $S(Z_{ij})$ becomes binary, the variable $Z_{ij}$ will not be updated in any subsequent iteration. Thus, we have to be careful about the initialization of $Z$. For example, if $Z$ is initialized so that $S(Z)$ is a binary matrix then the gate will not be trained. To ensure that the gate is trainable, we initialize each $Z_{ij}$ so that $0 < S(Z_{ij}) < 1$. This way, the $Z_{ij}$'s can have nonzero gradients at the start of optimization.

**Accelerating Convergence to Binary Solutions:** Recall that we need $S(Z)$ to converge to a binary matrix, in order for the gate $\tilde{q}$ to respect the cardinality constraint (i.e., to select at most $k$ experts). Empirically, we observe that if the optimizer runs for a sufficiently large number of iterations, then $S(Z)$ typically converges to a binary matrix. However, early stopping of the optimizer can be desired in practice for computational and statistical considerations, and this can prevent $S(Z)$ from converging. To encourage faster convergence towards a binary $S(Z)$, we will add an entropy regularizer to Problem (7). The following proposition is needed before we introduce the regularizer.

**Proposition 2.** *For any $z \in \mathbb{R}^m$, $\alpha \in \mathbb{R}^k$, and $Z \in \mathbb{R}^{k \times m}$, $r(S(z))$ and $\tilde{q}(\alpha, Z)$ belong to the probability simplex.*

The proof of the proposition is in Appendix B.1. Proposition 2 implies that, during training, the output of each single expert selector used by $\tilde{q}(\alpha, Z)$, i.e., $r(S(z^{(i)}))$ for $i \in [k]$, belongs to the probability simplex. Note that the entropy of each $r(S(z^{(i)}))$ is minimized by any one-hot encoded vector. Thus, for each $r(S(z^{(i)}))$, we add an entropy regularization term that encourages convergence towards one-hot encoded vectors; equivalently, this encourages convergence towards a binary $S(Z)$. Specifically, we solve the following regularized variant of Problem (7):

$$\min_{f_1,\dots f_n,\alpha,Z} \quad \sum_{(x,y)\in\mathcal{D}} \frac{1}{N}\ell\Big(y, \sum_{i=1}^{n} f_i(x)\tilde{q}(\alpha, Z)_i\Big) + \lambda\Omega(Z)$$

where $\Omega(Z) := \sum_{i=1}^{k} h\big(r(S(z^{(i)}))\big)$ and $h(.)$ is the entropy function. The hyperparameter $\lambda$ is non-negative and controls how fast each selector converges to a one-hot encoding. In our experiments, we tune over a range of $\lambda$ values. When selecting the best hyperparameters from tuning, we disregard any $\lambda$ whose corresponding solution does not have a binary $S(Z)$. In Appendix C.3, we report the number of training steps required for $S(Z)$ to converge to a binary matrix, on several real datasets. Other alternatives to ensure that $S(Z)$ converges to a binary matrix are also possible. One alternative is to regularize the entropy of each entry in $S(Z)$ separately. Another alternative is to anneal the parameter $\gamma$ of the smooth-step function towards zero.

**Softmax-based Alternative to Binary Encoding:** Recall that our proposed selectors in (6), i.e., $r\big(S(z^{(i)})\big), i \in [k]$, learn one-hot vectors *exactly* (by using binary encoding). One practical alternative for learning a one-hot vector is by using a softmax function with temperature annealing. Theoretically, this alternative cannot return a one-hot vector, but after training, the softmax output can be transformed to a one-hot vector using a heuristic (e.g., by taking an argmax). In Appendix C.1, we perform an ablation study in which we replace the selectors in DSelect-k with softmax functions (along with temperature annealing or entropy regularization).

## 3.2 DSelect-k for Per-example Gating

In this section, we generalize the static version of DSelect-k, $\tilde{q}(\alpha, Z)$, to the per-example gating setting. The key idea is to make the gate's parameters $\alpha$ and $Z$ functions of the input, so that the gate can make decisions on a per-example basis. Note that many functional forms are possible for these parameters. For simplicity and based on our experiments, we choose to make $\alpha$ and $Z$ linear functions of the input example. More formally, let $G \in \mathbb{R}^{k \times p}$, $W^{(i)} \in \mathbb{R}^{m \times p}$, $i \in [k]$, be a set of learnable parameters. Given an input example $x \in \mathbb{R}^p$, we set $\alpha = Gx$ and $z^{(i)} = W^{(i)}x$ in $\tilde{q}(\alpha, Z)$ (to simplify the presentation, we do not include bias terms). Thus, the per-example version of DSelect-k is a function $v$ defined as follows:

$$v(G, W, x) = \sum_{i=1}^{k} \sigma(Gx)_i r\big(S(W^{(i)}x)\big).$$

In the above, the term $r\big(S(W^{(i)}x)\big)$ represents the $i$-th single expert selector, whose output depends on the example $x$; thus different examples are free to select different experts. The term $\sigma(Gx)_i$ determines the input-dependent weight assigned to the $i$-th selector. The gate $v(G, W, x)$ is continuously differentiable in the parameters $G$ and $W$, so we propose optimizing it using first-order methods. Similar to the case of static gating, if $S(W^{(i)}x)$ is binary for all $i \in [k]$, then each $r\big(S(W^{(i)}x)\big)$ will select exactly one expert, and the example $x$ will be assigned to at most $k$ experts.

To encourage $S(W^{(i)}x)$, $i \in [k]$ to become binary, we introduce an entropy regularizer, similar in essence to that in static gating. However, unlike static gating, the regularizer here should be on a per-example basis, so that each example respects the cardinality constraint. By Proposition 2, for any $i \in [k]$, $r\big(S(W^{(i)}x)\big)$ belongs to the probability simplex. Thus, for each example $x$ in the training data, we introduce a regularization term of the form: $\Omega(W, x) := \sum_{i \in [k]} h\Big(r\big(S(W^{(i)}x)\big)\Big)$, and minimize the following objective function:

$$\sum_{(x,y) \in \mathcal{D}} \left( \frac{1}{N} \ell\Big(y, \sum_{i=1}^{n} f_i(x) v(G, W, x)_i\Big) + \lambda \Omega(W, x) \right),$$

where $\lambda$ is a non-negative hyperparameter. Similar to the case of static gating, we tune over a range of $\lambda$ values, and we only consider the choices of $\lambda$ that force the average number of selected experts per example to be less than or equal to $k$. If the application requires that the cardinality constraint be satisfied strictly for every example (not only on average), then annealing $\gamma$ in the smooth-step function towards zero enforces this.

## 4 Experiments

We study the performance of DSelect-k in the context of MTL and compare with state-of-the-art gates and baselines. In the rest of this section, we present experiments on the following real MTL datasets: MovieLens, Multi-MNIST, Multi-Fashion MNIST, and on a real-world, large-scale recommender system. Moreover, in Appendix C, we present an additional experiment on synthetic data (with up to 128 tasks), in which we study statistical performance and perform ablation studies.

**Competing Methods:** We focus on a multi-gate MoE, and study the DSelect-k and Top-k gates in both the static and per-example gating settings. For static gating, we also consider a Gumbel-softmax based gate [33]–unlike DSelect-k this gate cannot control the sparsity level explicitly (see the supplementary for details). In addition, we consider two MTL baselines. The first baseline is a MoE with a softmax gate (which uses all the available experts). The second is a *shared bottom* model [4], where all tasks share the same bottom layers, which are in turn connected to task-specific neural nets.

**Experimental Setup:** All competing models were implemented in TensorFlow 2. We used Adam [18] and Adagrad [7] for optimization, and we tuned the key hyperparameters using random grid search (with an average of 5 trials per grid point). Full details on the setup are in Appendix D.

### 4.1 MovieLens

**Dataset:** MovieLens [10] is a movie recommendation dataset containing records for 4,000 movies and 6,000 users. Following [36], for every user-movie pair, we construct two tasks. Task 1 is a binary classification problem for predicting whether the user will watch a particular movie. Task 2

is a regression problem to predict the user's rating (in $\{1, 2, \ldots, 5\}$) for a given movie. We use $1.6$ million examples for training and $200,000$ for each of the validation and testing sets.

**Experimental Details:** We use the cross-entropy and squared error losses for tasks 1 and 2, respectively. We optimize a weighted average of the two losses, i.e., the final loss function is $\alpha(\text{Loss of Task 1}) + (1 - \alpha)(\text{Loss of Task 2})$, and we report the results for $\alpha \in \{0.1, 0.5, 0.9\}$. The same loss function is also used for tuning and testing. The architecture consists of a multi-gate MoE with $8$ experts, where each of the experts and the task-specific networks is composed of ReLU-activated dense layers. For each $\alpha$, we tune over the optimization and gate-specific hyperparameters, including the number of experts to select (i.e., k in DSelect-k and Top-k). After tuning, we train each model for $100$ repetitions (using random initialization) and report the averaged results. For full details, see Appendix D.1.

**Results:** In Table 1, we report the test loss and the average number of selected experts. The results indicate that for all values of $\alpha$, either one of our DSelect-k gates (static or per-example) outperforms the competing methods, in terms of both the test loss and the number of selected experts. In the static gating setting, there does not seem to be a clear winner among the three competing methods (Top-k, DSelect-k, and Gumbel Softmax), but we note that DSelect-k outperforms both Top-k and Gumbel Softmax for two out of the three choices of $\alpha$. Notably, the softmax MoE is uniformly outperformed by the DSelect-k and Top-k gates, so sparsity in gating seems to be beneficial on this dataset. Our hypothesis is that softmax MoE is overfitting and the sparse gating methods are mitigating this issue. In Table C.2 in the appendix, we additionally report the individual task metrics (loss and accuracy).

| | | $\alpha = 0.1$ | | $\alpha = 0.5$ | | $\alpha = 0.9$ | |
| --- | --- | --- | --- | --- | --- | --- | --- |
| | | Loss | Experts | Loss | Experts | Loss | Experts |
| Static | DSelect-k | $4015 \pm 5$ | 2.7 | **$3804 \pm 3$** | **1.5** | $3690 \pm 2$ | **1.3** |
| | Top-k | **$4012 \pm 4$** | **2.0** | $3818 \pm 2$ | 2.0 | $3693 \pm 6$ | 2.0 |
| | Gumbel Softmax | $4171 \pm 3$ | 2.7 | $3898 \pm 2$ | 2.6 | **$3688 \pm 4$** | 3.6 |
| Per-example | DSelect-k | **$4006 \pm 6$** | **1.5** | $3823 \pm 3$ | **1.2** | $3679 \pm 2$ | **1.1** |
| | Top-k | $4027 \pm 8$ | 2.0 | $3841 \pm 4$ | 2.0 | $3741 \pm 3$ | 2.0 |
| Baselines | Softmax MoE | $4090 \pm 1$ | 8.0 | $3960 \pm 3$ | 8.0 | $3847 \pm 10$ | 8.0 |
| | Shared Bottom | $4037 \pm 2$ | - | $3868 \pm 2$ | - | $3687 \pm 1$ | - |

Table 1: Test loss (with standard error) and average number of selected experts on MovieLens. The parameter $\alpha$ is the weight of Task 1's loss (see text for details). The test loss is multiplied by $10^4$.

## 4.2 Multi-MNIST and Multi-Fashion MNIST

**Datasets:** We consider two image classification datasets: Multi-MNIST and Multi-Fashion [31], which are multi-task variants of the MNIST [20] and Fashion MNIST [37] datasets. We construct the Multi-MNIST dataset similar to [31]: uniformly sample two images from MNIST and overlay them on top of each other, and (ii) shift one digit towards the top-left corner and the other digit towards the bottom-right corner (by $4$ pixels in each direction). This procedure leads to $36 \times 36$ images with some overlap between the digits. The Multi-Fashion is constructed in a similar way by overlaying images from the Fashion MNIST dataset. For each dataset, we consider two classification tasks: Task 1 is to classify the top-left item and Task 2 is to classify the bottom-right item. We use 100,000 examples for training, and 20,000 examples for each of the validation and testing sets.

**Experimental Details:** We use cross-entropy loss for each task and optimize the sum of the losses[5]. The model is a multi-gate MoE with $8$ experts, where each expert is a convolutional neural network and each task-specific network is composed of a number of dense layers. We tune the optimization and gate-specific hyperparameters, including the number of experts to select, and use the average of the task accuracies as the tuning metric. After tuning, we train each model for $100$ repetitions (using random initialization) and report the averaged results. For full details, see Appendix D.2.

**Results:** In Table 2, we report the test accuracy and the number of selected experts for the Multi-MNIST and Multi-Fashion datasets. On Multi-MNIST, DSelect-k (static) outperforms Top-k and Gumbel Softmax, in terms of both task accuracies and number of selected experts. For example, it achieves over $1\%$ improvement in Task 2's accuracy compared to Top-k (static). DSelect-k (static) comes close to the performance of the Softmax MoE, but uses less experts (1.7 vs. $8$ experts). Here DSelect-k (per-example) does not offer improvement over the static variant (unlike the MovieLens dataset). On Multi-Fashion, we again see that DSelect-k (static) performs best in terms of accuracy.

---

[5]Due to the symmetry in the problem, assigning the two tasks equal weights is a reasonable choice.

|  |  | Multi-MNIST | | | Multi-Fashion MNIST | | |
|---|---|---|---|---|---|---|---|
|  |  | Accuracy 1 | Accuracy 2 | Experts | Accuracy 1 | Accuracy 2 | Experts |
| Static | DSelect-k | **92.56** ± 0.03 | **90.98** ± 0.04 | **1.7** | **83.78** ± 0.05 | **83.34** ± 0.05 | 1.8 |
|  | Top-k | 91.93 ± 0.06 | 90.03 ± 0.08 | 4 | 83.44 ± 0.07 | 82.66 ± 0.08 | 4 |
|  | Gumbel Softmax | 92.3 ± 0.05 | 90.63 ± 0.05 | 1.8 | 83.66 ± 0.07 | 83.28 ± 0.05 | **1.5** |
| Per-example | DSelect-k | **92.42** ± 0.03 | **90.7** ± 0.03 | **1.5** | **83.69** ± 0.04 | 83.13 ± 0.04 | **1.5** |
|  | Top-k | 92.27 ± 0.03 | 90.45 ± 0.03 | 4 | 83.66 ± 0.04 | **83.15** ± 0.04 | 4 |
| Baselines | Softmax MoE | 92.61 ± 0.03 | 91.0 ± 0.03 | 8 | 83.48 ± 0.04 | 82.81 ± 0.04 | 8 |
|  | Shared Bottom | 91.3 ± 0.04 | 89.47 ± 0.04 | - | 82.05 ± 0.05 | 81.37 ± 0.06 | - |

Table 2: Test accuracy (with standard error) and number of selected experts on Multi-MNIST/Fashion.

| Tasks / Methods | DSelect-k | Top-k |
|---|---|---|
| E. Task 1 (AUC) | **0.8103** ± 0.0002 | 0.7481 ± 0.0198 |
| E. Task 2 (AUC) | **0.8161** ± 0.0002 | 0.7624 ± 0.0169 |
| E. Task 3 (RMSE) | **0.2874** ± 0.0002 | 0.3406 ± 0.0180 |
| E. Task 4 (RMSE) | **0.8781** ± 0.0014 | 1.1213 ± 0.0842 |
| E. Task 5 (AUC) | **0.7524** ± 0.0003 | 0.5966 ± 0.0529 |
| S. Task 1 (AUC) | **0.6133** ± 0.0066 | 0.5080 ± 0.0047 |
| S. Task 2 (AUC) | **0.8468** ± 0.0289 | 0.5981 ± 0.0616 |
| S. Task 3 (AUC) | **0.9259** ± 0.0008 | 0.6665 ± 0.0091 |

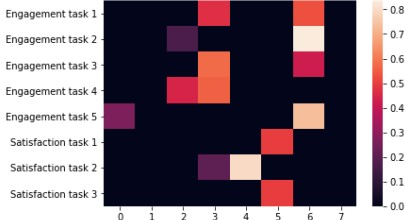

Table 3: Average performance (AUC and RMSE) and standard error on a real-world recommender system with 8 tasks: "E." and "S." denote engagement and satisfaction tasks, respectively.

Figure 3: Expert weights of the DSelect-k gates on the recommender system.

## 4.3   A Large-scale Recommender System

We study the performance of DSelect-k and Top-k in a real-word, large-scale content recommendation system. The system encompasses hundreds of millions of unique items and billions of users.

**Architecture and Dataset:** The system consists of a candidate generator followed by a multi-task ranking model, and it adopts a framework similar to [40, 34]. The ranking model makes predictions for 6 classification and 2 regression tasks. These can be classified into two categories: (i) engagement tasks (e.g., predicting user clicks, bad clicks, engagement time), and (ii) satisfaction tasks (e.g., predicting user satisfaction behaviors such as likes and dislikes). We construct the dataset from the system's user logs (which contain historical information about the user and labels for the 8 tasks). The dataset consists of billions of examples (we do not report the exact number for confidentiality). We use a random 90/10 split for the training and evaluation sets.

**Experimental Details:** We use the cross-entropy and squared error losses for the classification and regression tasks, respectively. The ranking model is based on a multi-gate MoE, in which each task uses a separate static gate. The MoE uses 8 experts, each composed of dense layers. For both the DSelect-k and Top-k based models, we tune the learning rate and the experts' architecture. Then, using the best hyperparameters, we train the final models for 5 repetitions (using random initialization). For additional details, see Appendix D.3.

**Results:** In Table 3, we report the out-of-sample performance metrics for the 8 tasks. The results indicate that DSelect-k outperforms Top-k on all tasks, with the improvements being most pronounced on the satisfaction tasks. In Figure 3, we show a heatmap of the expert weights chosen by the DSelect-k gates. Notably, for DSelect-k, all engagement tasks share at least one expert, and two of the satisfaction tasks share the same expert.

## 5   Conclusion

We introduced DSelect-k: a continuously differentiable and sparse gate for MoE, which can be trained using first-order methods. Given a user-specified parameter $k$, the gate selects at most $k$ of the $n$ experts. Such direct control over the sparsity level is typically handled in the literature by adding a cardinality constraint to the optimization problem. One of the key ideas we introduced is a binary encoding scheme that allows for selecting $k$ experts, without requiring any constraints in the optimization problem. We studied the performance of DSelect-k in MTL settings, on both synthetic and real datasets. Our experiments indicate that DSelect-k can achieve significant improvements in prediction and expert selection, compared to state-of-the-art MoE gates and MTL baselines.

**Societal Impact:** MoE models are used in various applications (as discussed in the introduction). DSelect-k can improve the interpretability and efficiency of MoE models, thus benefiting the underlying applications. We do not see direct, negative societal impacts from our proposal.

**Acknowledgements:** The research was conducted while Hussein Hazimeh was at Google, and part of the writing was done during his time at MIT. At MIT, Hussein Hazimeh and Rahul Mazumder acknowledge research funding from the Office of Naval Research [Grant ONR-N000141812298].

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
