# Supplementary to "DSelect-k: Differentiable Selection in the Mixture of Experts with Applications to Multi-Task Learning"

**Hussein Hazimeh[1], Zhe Zhao[1], Aakanksha Chowdhery[1], Maheswaran Sathiamoorthy[1]**

**Yihua Chen[1], Rahul Mazumder[2], Lichan Hong[1], Ed H. Chi[1]**

[1]Google, {hazimeh,zhezhao,chowdhery,nlogn,yhchen,lichan,edchi}@google.com
[2]Massachusetts Institute of Technology, rahulmaz@mit.edu

## A  Additional Related Work

**MTL:**  In MTL, deep learning-based architectures that perform soft-parameter sharing, i.e., share model parameters partially, are proving to be effective at exploiting both the commonalities and differences among tasks [6]. One flexible architecture for soft-parameter sharing is the multi-gate MoE [3].  We use the multi-gate MoE in our experiments and compare both sparse and dense gates—[3] considered only dense gates. In addition, several works have recently considered gate-like structures for flexible parameter sharing in MTL. For instance, [7, 4] give each task the flexibility to use or ignore components inside the neural network. The decisions are modeled using binary random variables, and the corresponding probability distributions are learned using SGD and the Gumbel-softmax trick [2]. This approach is similar to static gating, but it does not support per-example gating. Moreover, the number of nonzeros cannot be directly controlled (in contrast to our gate). Our work is also related to [5] who introduced "routers" (similar to gates) that can choose which layers or components of layers to activate per-task. The routers in the latter work are not differentiable and require reinforcement learning.

## B  Methodology Details

### B.1  Proofs

#### B.1.1  Proof Proposition 1

Let $f = \{f_i\}_{i \in [n]}$. To prove equivalence, we need to establish the following two directions: (I) an optimal solution $(f^*, \alpha^*, Z^*)$ to Problem (5) can be used to construct a feasible solution $(f, w)$ to Problem (2) and both solutions have the same objective, and (II) an optimal solution $(f^*, w^*)$ to Problem (2) can be used to construct a feasible solution $(f, \alpha, Z)$ to Problem (5) and both solutions have the same objective. Direction (I) is trivial: the solution defined by $f = f^*$ and $w = q(\alpha^*, Z^*)$ is feasible for Problem (2) and has the same objective as $(f^*, \alpha^*, Z^*)$.

Next, we show Direction (II). Let $s^* = \|w^*\|_0$ and denote by $t_j$ the index of the $j$-th largest element in $w^*$, i.e., the nonzero entries in $w^*$ are $w^*_{t_1} > w^*_{t_2} > \cdots > w^*_{t_{s^*}}$. For every $i \in [s^*]$, set $z^{(i)}$ to the binary representation of $t_i - 1$. If $s^* < k$, then we set the remaining (unset) $z^{(i)}$'s as follows: for $i \in \{s^* + 1, s^* + 2, \ldots, k\}$ set $z^{(i)}$ to the binary representation of $t_{s^*} - 1$. By this construction, the nonzero indices selected by $r(z^{(i)})$, $i \in [k]$ are exactly the nonzero indices of $w^*$.

To construct $\alpha$, there are two cases to consider: (i) $s^* = k$ and (ii) $s^* < k$. If $s^* = k$, then set $\alpha_i = \log(w^*_{t_i})$ for $i \in [k]$.  Therefore, $\sigma(\alpha)_i = w^*_{t_i}$ for $i \in [k]$, and consequently $q(\alpha, Z) = w^*$.

Otherwise, if $s^* < k$, then set $\alpha_i = \log(w^*_{t_i})$ for $i \in [s^* - 1]$ and $\alpha_i = \log(w^*_{t_{s^*}}/(k - s^*))$ for $i \in [s^*, s^* + 1, \ldots, k]$. Thus, for $i \in [s^* - 1]$, we have $\sigma(\alpha)_i = w^*_{t_i}$, i.e., the weights of the nonzero indices $t_j$, $j \in [s^* - 1]$ in $q(\alpha, Z)$ are equal to those in $w^*$. The weight assigned to the nonzero index $t_{s^*}$ in $q(\alpha, Z)$ is: $\sum_{i \in [s^*, s^*+1, \ldots, k]} \sigma(\alpha)_i = \sum_{i \in [s^*, s^*+1, \ldots, k]} w^*_{t_{s^*}}/(k - s^*) = w^*_{t_{s^*}}$. Therefore, $q(\alpha, Z) = w^*$. In both (i) and (ii), we have $q(\alpha, Z) = w^*$, so the solution $(f^*, \alpha, Z)$ is feasible and has the same objective as $(f^*, w^*)$.

### B.1.2 Proof of Proposition 2

First, we will use induction to show that $r(S(z))$ belongs to the probability simplex. Specifically, we will prove that for any integer $t \geq 1$ and $z \in \mathbb{R}^t$, $r(S(z))$ belongs to the probability simplex.

Our base case is for $t = 1$. In this case, there is a single binary encoding variable $z_1 \in \mathbb{R}$ and 2 experts. The single expert selector $r(S(z_1))$ is defined as follows: $r(S(z_1))_1 = 1 - S(z_1)$ and $r(S(z_1))_2 = S(z_1)$. The latter two terms are non-negative and sum up to 1. Thus, $r(S(z_1))$ belongs to the probability simplex.

Our induction hypothesis is that for some $t \geq 1$ and any $z \in \mathbb{R}^t$, $r(S(z))$ belongs to the probability simplex. For the inductive step, we need to show that for any $v \in \mathbb{R}^{t+1}$, $r(S(v))$ belongs to the probability simplex. From the definition of $r(.)$, the following holds:

$$r(S(v))_i = \begin{cases} r(S([v_1, v_2, \ldots, v_t]^T))_i (1 - S(v_{t+1})) & i \in [2^t] \\ r(S([v_1, v_2, \ldots, v_t]^T))_{i-2^t} S(v_{t+1}) & i \in [2^{t+1}] \setminus [2^t] \end{cases} \tag{B.1}$$

By the induction hypothesis, we have $r(S([v_1, v_2, \ldots, v_t]^T))_i \geq 0$ for any $i$. Moreover, $S(.)$ is a non-negative function. Therefore, $r(S(v))_i \geq 0$ for any $i$. It remains to show that the sum of the entries in $r(S(v))$ is 1, which we establish next:

$$\sum_{i=1}^{2^{t+1}} r(S(v))_i = \sum_{i=1}^{2^t} r(S(v))_i + \sum_{i=2^t+1}^{2^{t+1}} r(S(v))_i$$

$$\overset{(B.1)}{=} \sum_{i=1}^{2^t} r(S([v_1, \ldots, v_t]^T))_i (1 - S(v_{t+1})) + \sum_{i=2^t+1}^{2^{t+1}} r(S([v_1, \ldots, v_t]^T))_{i-2^t} S(v_{t+1})$$

$$\tag{B.2}$$

Using a change of variable, the second summation in the above can be rewritten as follows: $\sum_{i=2^t+1}^{2^{t+1}} r(S([v_1, v_2, \ldots, v_t]^T))_{i-2^t} S(v_{t+1}) = \sum_{i=1}^{2^t} r(S([v_1, v_2, \ldots, v_t]^T))_i S(v_{t+1})$. Plugging the latter equality into (B.2) and simplifying, we get $\sum_{i=1}^{2^{t+1}} r(S(v))_i = 1$. Therefore, $r(S(v))$ belongs to the probability simplex, which establishes the inductive step. Finally, we note that $\tilde{q}(\alpha, Z)$ belongs to the probability simplex since it is a convex combination of probability vectors.

### B.2 Extending DSelect-k to arbitrary $n$

Suppose that the number of experts $n$ is not a power of 2. For the DSelect-k gate $\tilde{q}(\alpha, Z)$ to work in this setting, we need each single expert selector $r$ used by the gate to be able to handle $n$ experts. Next, we discuss how $r$ can handle $n$ when it is not a power of 2. Let $m$ be the smallest integer so that $n < 2^m$. Then, we treat the problem as if there are $2^m$ experts and use a binary encoding vector $z \in \mathbb{R}^m$. For $i \in [n]$, we let entry $r(z)_i$ be the weight of expert $i$ in the MoE. Note that the entries $r(z)_i$, $i \in \{n+1, n+2, \ldots, 2^m\}$, are not associated with any experts. To avoid a situation where $r(z)$ assigns nonzero probability to the latter entries, we add the following penalty to the objective function: $\frac{\xi}{\sum_{i \in [n]} r(z)_i}$ where $\xi$ is a non-negative parameter used to control the strength of the penalty. This penalty encourages to $r(z)_i$, $i \in [n]$ (i.e., the entries associated with the $n$ experts) to get more probability. In our experiments, we observe that $\sum_{i \in [n]} r(z)_i$ converges to 1, when $\xi$ is sufficiently large. The penalty described above is part of our TensorFlow implementation of DSelect-k. We also note that there are other potential alternatives to deal with the entries $r(z)_i$, $i \in \{n+1, n+2, \ldots, 2^m\}$, without adding a penalty to the objective function. For example, one alternative is to randomly assign each of $r(z)_i$, $i \in \{n+1, n+2, \ldots, 2^m\}$ to one of the $n$ experts.

### B.3 Smooth-step Function

In Figure B.1, we plot the smooth-step function [1] and the logistic function $L(x) = (1 + e^{-6x})^{-1}$. Note that the logistic function is re-scaled to be on the same scale as the smooth-step function.

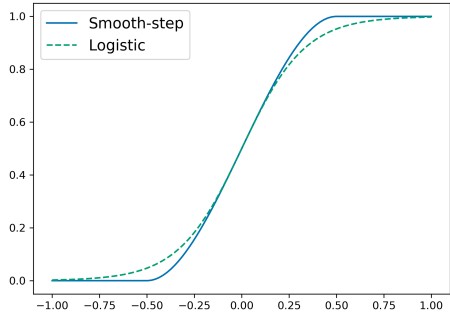

Figure B.1: The Smooth-step ($\gamma = 1$) and Logistic functions.

## C   Additional Experimental Results

### C.1   Prediction and Expert Selection Performance on Synthetic Data

In this experiment, we aim to (i) understand how the number of experts and tasks affects the prediction and expert selection performance for the different gates, and (ii) quantify the benefit from binary encoding in our gate through an ablation study. We focus on a static gating setting, where we consider the DSelect-k and Top-k gates, in addition to two variants of the DSelect-k gate used for ablation. To better quantify the expert selection performance and avoid model mis-specification, we use synthetic data generated from a multi-gate MoE. First, we describe our data generation process.

**Synthetic Data Generation:**   We consider $128$ regression tasks, separated into four mutually exclusive groups: $\{G_i\}_{i \in [4]}$, where $G_i$ is the set of indices of the tasks in group $i$. As we will discuss next, the tasks are constructed in a way so that tasks within each group are highly related, while tasks across groups are only marginally related. Such a construction mimics real-world applications in which tasks can be clustered in terms of relatedness.

Each group consists of 16 tasks which are generated from a group-specific MoE. The group-specific MoE consists of 4 experts: $\{f_i\}_{i \in [4]}$. Each expert is the sum of 4 ReLU-activated units. The output of each task in the group is a convex combination of the 4 experts. Specifically, for each task $t \in [16]$ in the group, let $\alpha^{(t)} \in \mathbb{R}^4$ be a task-specific weight vector. Then, given an input vector $x$, the output of task $t$ is defined as follows:

$$y^{(t)}(x) := \sum_{i=1}^{4} \sigma(\alpha^{(t)})_i f_i(x)$$

For each group, we create an instance of the group-specific MoE described above, where we initialize all the weights randomly and independently from the other groups. In particular, we sample the weights of each expert independently from a standard normal distribution. To encourage relatedness among tasks in each group, we sample the task weights $[\alpha^{(1)}, \alpha^{(2)} \dots \alpha^{(16)}]$ from a zero-mean multivariate normal distribution where we set the correlation between any two task weights to $0.8$.

To generate the data, we sample a data matrix $X$, with $140{,}000$ observations and $10$ features, from a standard normal distribution. The data matrix is shared by all $128$ tasks and the regression outputs are obtained by using $X$ as an input to each group-specific MoE. We use $100{,}000$ observations for the training set and $20{,}000$ observations for each of the validation and testing sets.

**Experiment Design:**   We consider a multi-gate MoE and compare the following static gates: DSelect-k gate, Top-k gate, and an "ablation" gate (which will be discussed later in this section). Our goal is to study how, for each gate, the number of tasks affects the prediction and expert selection

performance. To this end, we consider $4$ regression problems, each for a different subset of the $128$ tasks; specifically, we consider predicting the tasks in (i) $G_1$ (16 tasks), (ii) $G_1 \cup G_2$ (32 tasks), (iii) $G_1 \cup G_2 \cup G_3$ (64 tasks), and (iv) $G_1 \cup G_2 \cup G_3 \cup G_4$ (128 tasks). In each of the four problems, we use a multi-gate MoE to predict the outputs of the corresponding tasks simultaneously. The MoE has the same number of experts used to generate the data, i.e., if $T$ is the total number of tasks in the problem, the MoE consists of $T/4$ experts, where the experts are similar to those used in data generation (but are trainable). Each task is associated with a task-specific gate, which chooses a convex combination of $4$ out of the $T/4$ experts. Note that unlike the architecture used to generate the data, each task gate here is connected to all experts, even those belonging to the unrelated groups. The architecture used to generate the data can be recovered if the task gates across groups do not share experts, and the task gates within each group share the same $4$ experts. We use squared error loss for training and tuning.

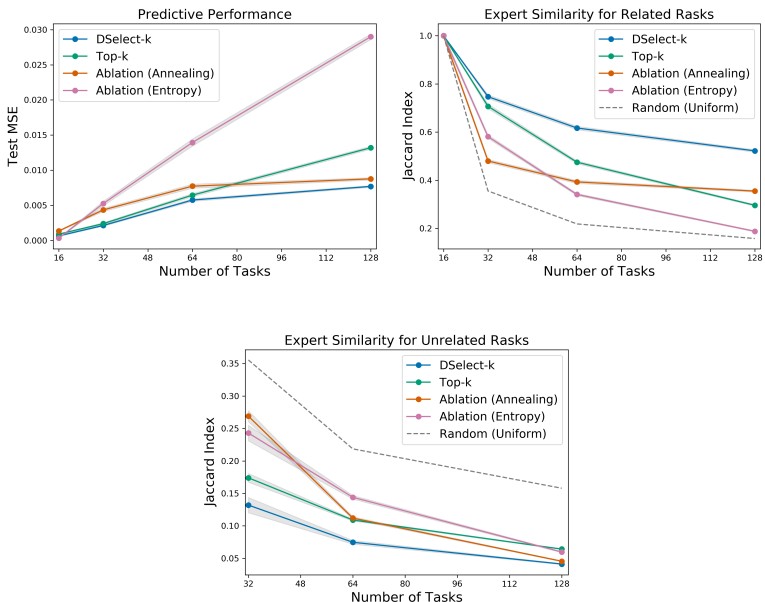

Figure C.2: Predictive and expert selection performance on synthetic data generated from a MoE.

**Ablation:** In addition to comparing the DSelect-k and Top-k gates, we perform an ablation study to gain insight on the role of binary encoding in the DSelect-k gate. Recall that in the DSelect-k gate, we introduced the single expert selector which learns a one-hot encoded vector (using a binary encoding scheme). In the literature, a popular way to learn such one-hot encoded vectors is by using a softmax (with additional heuristics such as temperature annealing to ensure that the probability concentrates on one entry). Thus, in our ablation study, we consider the DSelect-k gate $\tilde{q}(\alpha, Z)$, and we replace each single expert selector $r(.)$ with a softmax-based selector. More precisely, let $\alpha \in \mathbb{R}^k$ and $\beta^{(i)} \in \mathbb{R}^n$, $i \in [k]$, be learnable parameter vectors. Then, we consider the following "ablation" gate: $h(\alpha, \beta) := \sum_{i=1}^{k} \sigma(\alpha)_i \sigma(\beta^{(i)})$. Here, $\sigma(\alpha)_i$ determines the weight assigned to selector $i$, and $\sigma(\beta^{(i)})$ acts as a surrogate to the single expert selector $r(S(z^{(i)}))$. To ensure that $\sigma(\beta^{(i)})$ selects a single expert (i.e., leads to a one-hot encoding), we consider two alternatives: (i) annealing the temperature of $\sigma(\beta^{(i)})$ during training[1], and (ii) augmenting the objective with an entropy regularization term (similar to that of the DSelect-k gate) to minimize the entropy of each $\sigma(\beta^{(i)})$. In our results, we refer to (i) by "Ablation (Annealing)" and to (ii) by "Ablation (Entropy)". Note that these two ablation alternatives can converge to a one-hot encoding asymptotically (due to the nature of the softmax), whereas our proposed gate can converge in a finite number of steps.

---

[1]There are pathological cases where annealing the temperature in softmax will converge to more than one nonzero entry. This can happen when multiple entries in the input to the softmax have exactly the same value.

**Measuring Expert Selection Performance:** To quantify the similarity between the experts selected by the different tasks, we use the Jaccard index. Given two tasks, let $A$ and $B$ be the sets of experts selected by the first and second tasks, respectively. The Jaccard index of these two sets is defined by: $|A \cap B|/|A \cup B|$. In our experiments, we compute: (i) the average Jaccard index for the related tasks, and (ii) the average Jaccard index for unrelated tasks. Specifically, we obtain (i) by computing the Jaccard index for each pair of related tasks, and then averaging. We obtain (ii) by computing the Jaccard index over all pairs of tasks that belong to different groups (i.e., pairs in the same group are ignored), and then averaging.

**Results:** After tuning, we train each competing model, with the best hyperparameters, for 100 randomly-initialized repetitions. In Figure C.2, we plot the performance measures (averaged over the repetitions) versus the number of tasks. In the left plot, we report the MSE on the test set. In the middle and right plots we report the (averaged) Jaccard index for the related and unrelated tasks, respectively. In the latter two plots, we also consider a random gate which chooses 4 experts uniformly at random, and plot the expected value of its Jaccard index. In Figure C.2 (middle), a larger index is better since the related tasks will be sharing more experts. In contrast, in Figure C.2 (right), a lower index is preferred since the unrelated tasks will be sharing less experts. For all methods, the Jaccard index in Figures C.2 (middle) and (right) decreases with the number of tasks. This is intuitive, since as the number of tasks increases, we use more experts, giving any two given gates more flexibility in choosing mutually exclusive subsets of experts.

Overall, the results indicate that DSelect-k gate significantly outperforms Top-k in all the considered performance measures, and the differences become more pronounced as the number of tasks increases. For example, at 128 tasks, DSelect-k achieves over $40\%$ improvement in MSE and $76\%$ improvement in Jaccard index for related tasks, compared to Top-k. The DSelect-k gate also outperforms the two ablation gates in which we replace the binary encoding by a Softmax-based selector. The latter improvement suggests that the proposed binary encoding scheme is relatively effective at selecting the right experts. We also investigated the poor performance of the Ablation (Entropy) gate, and it turns out that the Softmax-based single expert selectors, i.e., the $\sigma(\beta^{(i)})$'s, tend to select the same expert. Specifically, we set $k = 4$ in the ablation gate, but it ends up selecting $\sim 2$ experts in many of the training repetitions. In contrast, the DSelect-k and Top-k gates select 4 experts.

## C.2 Gate Visualizations

### C.2.1 MovieLens

In Figure C.3, we plot the expert weights during training on the MovieLens dataset, for the Top-k and DSelect-k gates (after tuning both models). The plots show that Top-k exhibits frequent "jumps", where in a single training step an expert's weight can abruptly change from a nonzero value to zero. These jumps keep occurring till the end of training (at around $10^5$ training steps). In contrast, DSelect-k has smooth transitions during training. Additional details on the MovieLens dataset and the MoE architecture used can be found in Section 4.1 of the paper.

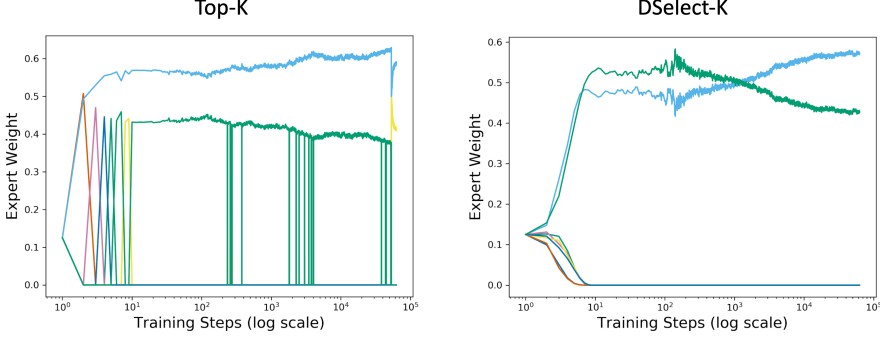

Figure C.3: Expert weights during training on the MovieLens dataset. Each color corresponds to a separate expert. The plots are for the best models obtained after tuning.

### C.2.2 Synthetic Data

Here we consider a binary classification dataset generated from a static MoE that consists of 4 experts. We train another MoE model which employs 16 experts: 4 of these experts are copies (i.e., have exactly same weights) of the 4 experts used in data generation, and the rest of the experts are randomly initialized. We freeze all the experts and train only over the gate parameters. In this simple setting, we expect the gate to be able to recover the 4 experts that were used to generate the data. We trained two MoE models: one based on Top-k and another based on DSelect-k. After tuning both models, Top-k recovered only 1 right expert (and made 3 mistakes), whereas our model recovered all 4 experts. In Figures C.4 and C.5, we plot the expert weights during training, for the Top-k and DSelect-k gates, respectively. The Top-k exhibits a sharp oscillatory behavior during training, whereas DSelect-k has smooth transitions.

**Additional details on data generation and model:**  We consider a randomly initialized "data-generating" MoE with 4 experts (each is a ReLU-activated dense layer with 4 units). The output of the MoE is obtained by taking the average of the 4 experts and feeding that into a single logistic unit. We generate a multivariate normal data matrix $X$ with 20,000 observations and 10 features (10,000 observations are allocated for each of the training and validation sets) . To generate binary classification labels, we use $X$ as an input to the data-generating MoE and apply a sign function to the corresponding output. For training and tuning, we consider a MoE architecture with 16 experts: 4 of these experts are copies of the experts used in data generation, and the rest of the experts are initialized randomly. All experts are frozen (not trainable). A trainable gate chooses 4 out of the 16 experts, and the final result is fed into a single logistic unit. We optimized the cross-entropy loss using Adam with a batch size of 256, and tuned over the learning rate in $\{10^{-1}, 10^{-2}, \ldots, 10^{-5}\}$.

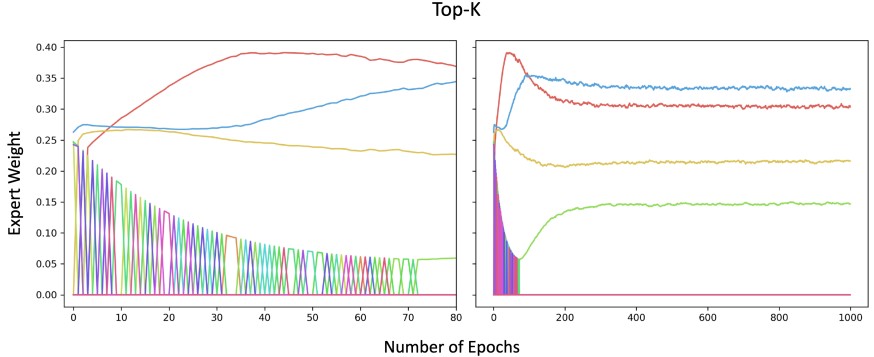

Figure C.4: Expert weights during training on synthetic data generated from a MoE. Each color corresponds to a separate expert. The left plot is a magnified version of the right plot. The plots are for the best model obtained after tuning.

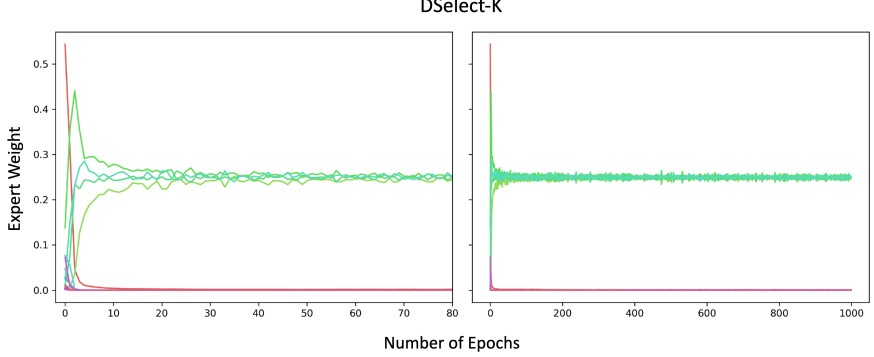

Figure C.5: Expert weights during training on synthetic data generated from a MoE. Each color corresponds to a separate expert. The left plot is a magnified version of the right plot. The plots are for the best model obtained after tuning.

## C.3 Gate Convergence and FLOPS

In Table C.1, we report the percentage of training steps required for $S(Z)$ to converge to a binary matrix in the DSelect-k gate, on several real datasets. These results are based on the tuned models discussed in Section 4 of the paper. We also report the number of floating point operations (FLOPS) required by the MoE based on DSelect-k relative to the MoE based on Top-k, during training. The results indicate that the number of training steps till convergence to a binary matrix depends on the specific dataset: ranging from only $0.04\%$ on the MovieLens dataset to $80\%$ on the Multi-Fashion MNIST. Moreover, on certain datasets (MovieLens with $\alpha = 0.9$ and Multi-MNIST), DSelect-k requires less FLOPS during training than Top-k, i.e., DSelect-k is effective at conditional training (on these particular datasets).

Table C.1: We report two statistics: (i) Percentage of training steps required for the DSelect-k gate to converge to a binary matrix, and (ii) the number of FLOPS needed by the DSelect-k based MoE during training relative to that of Top-k. The parameter $\alpha$ controls the weight assigned to task 1's loss in the MovieLens dataset—see Section 4.1 of the paper for more details.

| | MovieLens | | | Multi-MNIST | Multi-Fashion |
|---|---|---|---|---|---|
| | $\alpha = 0.1$ | $\alpha = 0.5$ | $\alpha = 0.9$ | | |
| % Training Steps until Binary $S(Z)$ | 11.37 | 9.39 | 0.04 | 42.33 | 80.02 |
| FLOPS(DSelect-k)/FLOPS(Top-k) | 1.5 | 1.2 | 0.6 | 0.8 | 1.2 |

## C.4 MovieLens

In Table C.2, we report the accuracy for task 1 (classification) and the loss for task 2 ( regression) for the competing methods on the MovieLens dataset.

Table C.2: Mean test loss for task 2 (T2) and accuracy for task 1 (T1) on MovieLens (standard error is shown next to each mean). The parameter $\alpha$ determines the weight of Task 1's loss (see text for details). The test loss is multiplied by $10^4$.

| | | $\alpha = 0.1$ | | $\alpha = 0.5$ | | $\alpha = 0.9$ | |
|---|---|---|---|---|---|---|---|
| | | T2 Loss | T1 Accuracy | T2 Loss | T1 Accuracy | T2 Loss | T1 Accuracy |
| Static | DSelect-k | $4038 \pm 5$ | $83.03 \pm 0.07$ | $3926 \pm 6$ | $83.86 \pm 0.02$ | $3943 \pm 5$ | $84.04 \pm 0.01$ |
| | Top-k | $4056 \pm 4$ | $84.09 \pm 0.05$ | $4002 \pm 4$ | $84.21 \pm 0.02$ | $3884 \pm 4$ | $84.17 \pm 0.02$ |
| | Gumbel Softmax | $4172 \pm 2$ | $80.76 \pm 0.04$ | $4085 \pm 3$ | $83.71 \pm 0.01$ | $3878 \pm 4$ | $84.17 \pm 0.02$ |
| Per-example | DSelect-k | $4030 \pm 7$ | $83.03 \pm 0.12$ | $3981 \pm 7$ | $83.92 \pm 0.03$ | $3932 \pm 1$ | $84.07 \pm 0.02$ |
| | Top-k | $4057 \pm 9$ | $83.28 \pm 0.1$ | $3995 \pm 8$ | $83.91 \pm 0.03$ | $3914 \pm 4$ | $84.05 \pm 0.01$ |
| Baselines | Softmax MoE | $4047 \pm 1$ | $78.73 \pm 0.01$ | $4028 \pm 3$ | $83.56 \pm 0.01$ | $3970 \pm 3$ | $83.86 \pm 0.02$ |
| | Shared Bottom | $3993 \pm 2$ | $79.06 \pm 0.02$ | $3875 \pm 2$ | $82.65 \pm 0.02$ | $3991 \pm 2$ | $84.01 \pm 0.01$ |

# D   Experimental Details

**Computing Setup:** We ran the experiments on a cluster that automatically allocates the computing resources. We do not report the exact specifications of the cluster for confidentiality.

**Gumbel-softmax Gate:** [7, 4] present an approach for learning which layers in a neural network to activate on a per-task basis. The decision to select each layer is modeled using a binary random variable whose distribution is learned using the Gumbel-softmax trick. Note that the latter approach does not consider a MoE model. Here we adapt the latter approach to the MoE; specifically we consider a Gumbel-softmax gate that uses binary variables to determine which experts to select. Given $n$ experts $\{f_i\}_{i=1}^n$, this gate uses $n$ binary random variables $\{U_i\}_{i=1}^n$, where $U_i$ determines whether expert $f_i$ is selected. Moreover, the gate uses an additional learnable vector $\alpha \in \mathbb{R}^n$ that determines the weights of the experts. Specifically, the gate is a function $d(\alpha, U)$ whose $i$-th component (for any $i \in [n]$) is given by:

$$d(\alpha, U)_i = \sigma(\alpha)_i U_i$$

To learn the distribution of the $U_i$'s, we use the Gumbel-softmax trick as described in [7]. Moreover, following [7], we add the following sparsity-inducing penalty to the objective function: $\lambda \sum_{i \in [n]} \log \psi_i$, where $\psi_i$ is the Bernoulli distribution parameter of $U_i$, and $\lambda$ is a non-negative parameter used to control the number of nonzeros selected by the gate. Note that the latter penalty cannot directly control the number of nonzeros as in DSelect-k or Top-k.

### D.1 MovieLens

**Architecture:** For MoE-based models, we consider a multi-gate MoE architecture (see Figure 1), where each task is associated with a separate gate. The MoE uses $8$ experts, each of which is a ReLU-activated dense layer with $256$ units, followed by a dropout layer (with a dropout rate of $0.5$). For each of the two tasks, the corresponding convex combination of the experts is fed into a task-specific subnetwork. The subnetwork is composed of a dense layer (ReLU-activated with $256$ units) followed by a single unit that generates the final output of the task. The shared bottom model uses a dense layer (whose number of units is a hyperparameter) that is shared by the two tasks, followed by a dropout layer (with a rate of $0.5$). For each task, the output of the shared layer is fed into a task-specific subnetwork (same as that of the MoE-based models).

**Hyperparameters and Tuning:** We tuned each model using random grid search, with an average of $5$ trials per grid point. We used Adagrad with a batch size of $128$ and considered the following hyperparameters and ranges: Learning Rate: $\{0.001, 0.01, 0.1, 0.2, 0.3\}$, Epochs: $\{5, 10, 20, 30, 40, 50\}$, k for Top-k and DSelect-k: $\{2, 4\}$, $\lambda$ for DSelect-k: $\{0.1, 1, 10\}$, $\gamma$ for smooth-step: $\{1, 10\}$, Units in Shared bottom: $\{32, 256, 2048, 4096, 8192\}$, $\lambda$ in Gumbel-softmax: $\{10^{-6}, 10^{-5}, \dots, 10\}$. For Gumbel-softmax, we pick the best solution whose expected number of nonzeros is less than or equal to $4$.

### D.2 Multi-MNIST and Multi-Fashion MNIST

**Architecture:** MoE-based models use a multi-gate MoE (as in Figure 1). Each of the $8$ experts is a CNN that is composed (in order) of: (i) convolutional layer 1 (kernel size = 5, number of filters = 10, ReLU-activated) followed by max pooling, (ii) convolutional layer 2 (kernel size = 5, number of filters = 20, ReLU-activated) followed by max pooling, and (iii) a stack of ReLU-activated dense layers with $50$ units each (the number of layers is a hyperparameter). The subnetwork specific to each task is composed of a stack of 3 dense layers: the first two have 50 ReLU-activated units and the third has 10 units followed by a softmax. The shared bottom model uses a shared CNN (with the same architecture as the CNN in the MoE). For each task, the output of the shared CNN is fed into a task-specific subnetwork (same as that of the MoE-based models).

**Hyperparameters and Tuning:** We tuned each model using random grid search, with an average of $5$ trials per grid point. We used Adam with a batch size of $256$ and considered the following hyperparameters and ranges: Learning Rate: $\{0.01, 0.001, 0.0001, 0.00001\}$, Epochs: $\{25, 50, 75, 100\}$, k for Top-k and DSelect-k: $\{2, 4\}$, $\gamma$ for smooth-step: $\{0.1, 1, 10\}$, Number of dense layers in CNN: $\{1, 3, 5\}$, $\lambda$ in Gumbel-softmax: $\{0, 10^{-3}, 10^{-2}, 10^{-1}, 1, 10, 1000\}$. For Gumbel-softmax, we pick the best solution whose expected number of nonzeros is less than or equal to $4$.

### D.3 Recommender System

Each of the $8$ experts in the MoE consists of a stack of ReLU-activated dense layers with $256$ units each. We fix k to 2 in both DSelect-k and Top-k. We tune over the learning rate and architecture. For both models, training is terminated when there is no significant improvement in the validation loss.

### D.4 Synthetic Data

We tuned each model using random grid search, with an average of $5$ trials per grid point. We used Adam with a batch size of $256$ and considered the following hyperparameters and ranges: Learning rate: $\{0.001, 0.01, 0.1\}$, Epochs $\{25, 50, 75, 100\}$, $\gamma$ for smooth-step: $\{5, 10, 15\}$, $\lambda$ for DSelect-k: $\{0.001, 0.005, 0.01, 0.1\}$, $\lambda$ for Ablation (Entropy): $\{10^{-6}, 10^{-5}, 10^{-4}, 10^{-3}, 10^{-2}, 10^{-1}, 1, 100\}$. Moreover, for Ablation (Annealing), we anneal the temperature of softmax starting from a hyperparameter $s$ down to $10^{-16}$ (the temperatures are evenly spaced on a logarithmic scale). We tune the starting temperature $s$ over $\{10^{-6}, 10^{-5}, 5 \times 10^{-5}, 10^{-4}, 2.5 \times 10^{-4}, 5 \times 10^{-4}, 7.5 \times 10^{-4}, 10^{-3}, 5 \times 10^{-3}, 10^{-2}\}$ (note that such a fine grid was necessary to get annealing to work for the ablation gate).