# OpenReview forum: "DSelect-k: Differentiable Selection in the Mixture of Experts with Applications to Multi-Task Learning"
_NeurIPS.cc/2021/Conference — NeurIPS 2021 Poster_

### Official Review · Reviewer_GzAm · 2021-07-16

**Rating:** 9
**Confidence:** 4

**Summary:**

The authors propose a novel differentiable sparse gate for mixture-of-experts models.
At-most-k selection problem, a cardinality-constrained problem, is reformulated as an unconstrained problem using a binary encoding scheme.  A continuous relaxation of the binary variable can be optimized with first-order methods with careful initialization and regularization.
The proposed method outperforms conventional methods in multitask learning settings.

**Limitations And Societal Impact:**

No discussion about potential negative societal impact of their work are provided.  I cannot find one, too.

**Main Review:**

Originality:
The proposed method is novel for mixture-of-experts models.  The differences from previous work are clearly described.


Quality:
The proposed binary encoding scheme is reasonable, and the superiority of the differentiable sparse gate is supported by both the theoretical analysis the experimental results.


Clarity:
The paper is well organized and easy to understand.
I would like to know how W is initialized to be 0 < S(Wx) <1 for Per-example Gating.


Significance:
The proposed differentiable gating technique can not only be a baseline method to train mixture-of-experts models but also be applicable to other models, such as attention models.
If the experimental results on the same task as [32] are provided, the impact of this paper might be more significant.


**Time Spent Reviewing:**

4

---

> ### Author Response · Authors · 2021-08-10
> **Reply to Reviewer GzAm**
>
> Thank you for taking the time to review the paper and for your feedback.
>
> Regarding initialization in the per-example gating: We initialized $W^i$ by sampling from a zero-mean uniform distribution (we used the default uniform distribution in TensorFlow, which has a support of [-0.05, +0.05]). The expected value of $W^i x$ is the zero vector (assuming x is fixed or sampled independently from $W^i$). It can also be checked that the expected value of $S(W^i x)$ is the vector whose all entries are 0.5. But of course, there is no guarantee (in a deterministic way) that the entries of $S(W^i x)$ will be non-binary. Initialization here seems to be less problematic than the static setting: even if $S(W^i x)$ is initialized to binary values for some example x, training will still be possible if any other example in the batch is initialized to a non-binary value. Note: In case training issues arise due to initialization, the support of the uniform distribution can be tightened (or alternatively the features can be rescaled); however, for the datasets considered, we saw no need for this. Thank you for raising this point.

---

### Official Review · Reviewer_GSeM · 2021-07-17

**Rating:** 6
**Confidence:** 4

**Summary:**

The submission studies how to design a differentiable module for selecting K experts out of N experts in a sparse MoE model.
The proposed method reparametrize the softmax distribution over N the experts as log(N) binary variables.
It then proposes a smoothed S-shaped continuous approximation of the hard binary function.

**Limitations And Societal Impact:**

The authors are well aware that the proposed method may be slower than the Top-k method when it comes to training since it only supports conditional training partially due to the soft approximation. Please see Cons in Main Review for my extra concerns.

**Main Review:**

Pros:

- A novel differentiable N-select-K module that is built upon new ideas (the binary encoding and the S-shaped smoothed function).
Such a radically different alternative to the existing methods (Top-K, Gumbel-softmax variants, etc) is welcome since the existing methods are still hard to tune (based on my personal experience with sparse MoE).

- The writing is of high quality, and the source code is provided.

- It presents extensive empirical analysis and consistent improvement over the existing method.

Cons:

- The proposed smooth function S(t) may lead to dead experts (i.e., some experts will no longer be used after a certain number of training iterations) or may make the model converge to a bad local optimum, because the proposed function has no gradient once t is below -gamma/2 or above gamma/2.

- The proposed module can select the same expert for more than one time. In some applications, we may want to select k DISTINCT experts, which is not supported by the proposed module.

- As far as I know, most sparse MoE models suffer from the dead-expert issue (a significant number of experts are never selected after optimization). There is no analysis of how the proposed method performs when it comes to the dead-expert issue.



**Time Spent Reviewing:**

4

---

> ### Author Response · Authors · 2021-08-10
> **Reply to Reviewer GSeM**
>
> Thank you for taking the time to review the paper and for your feedback. Below is an (in-order) discussion on the cons mentioned in your review:
>
> 1. By the end of training, we purposefully want the smooth-step function to return binary values (where the gradient is zero), so that the gate selects at most k experts. We do not view convergence to binary values as an issue by itself. However, we do agree that getting good solutions/local minima can be tricky, given the nature of the smooth-step function. Specifically, the speed of convergence towards binary values can affect the quality of the final solution. That’s why we use entropy regularization and tune over the corresponding parameter to control the speed of convergence towards binary values.
>
> 2. We used this cardinality constrained formulation (<=k as opposed to exact k) because it gives the optimization algorithm the freedom to select <k experts if it deems that beneficial. For example, if solutions with k-1 and k experts achieve similar training losses, then the one with k-1 experts is usually preferred. Our formulation seems to be effective for multi-task learning, but we do agree that in other applications selecting exactly k experts/objects may be preferred. Please see also the reply to point 2 for reviewer XzFT.
>
> 3. Thank you for raising this point. We think that the effect of having dead experts depends on the application, dataset, and model architecture. There are cases where dead experts are desired. For example, if the number of experts is misspecified to be too large, then having dead experts (i.e., not selecting some experts at all) may be beneficial. In settings where dead experts are an issue, previous work (e.g., Shazeer et al., 2017 and Bengio et al., 2015) had suggested adding extra penalties to the objective function to balance out expert utilization--such penalties can also be used in our approach.

---

### Official Review · Reviewer_1DHt · 2021-07-18

**Rating:** 6
**Confidence:** 3

**Summary:**

This paper presents DSelect-k, a differentiable selection method in MoE. The DSelect-k is designed based on based on a novel binary encoding formulation, it requires to uses all the available experts at the early training stage, and is able to converge to a small subset of the experts so computation can be saved. The DSelect-k is mainly evaluated in the multi-task learning setting on datasets (e.g., in recommendation system).

**Ethical Concerns:**

I don't think there are any ethical issues with this paper.

**Limitations And Societal Impact:**

The authors adequately addressed the limitations and potential negative societal impact of their work.

**Main Review:**

* Originality:  MoE has been increasingly important in multi-task learning and large-scale model training. This work attempts to solve the discontinuous and non-smooth issue in the top-k operation in MoE by introducing a differentiable select-k method named DSeleck-k. The DSelect-k method is based on a novel binary encoding formulation. The reviewer want to point out another work about differentiable top-k [1], detailed comments can be found in Q2.

* Quality: The submission is technically sound. The binary encoding formulation is novel and interesting to me. The proposed method is evaluated in the multi-task learning setting (in MovieLens and MNIST).

* Clarity: The paper is well-written.

* Significance:  This work improves the MoE training. It achieves comparable or better performance in the context of multi-task learning. Another important application of MoE, large-scale pre-training, was not covered (See detailed comments in Q1).

I have the following detailed questions:

* Q1: As the author mentioned in related work, one of the most recent applications of MoE is in language model pre-training (like GShard and Switch Transformer), where the FFN in the Transformer model is replaced by a MoE layer to enlarge the number of parameters. E.g., the Switch Transformer has 1.6 trillion parameters. Could the authors discuss the possibility of applying DSelect-k in training huge models like GShard and Switch Transformer. My main concern is that, the proposed DSelect-k has to forward all the available experts at the early training stage, however it is super expensive to fully forward a trillion-parameter model.

* Q2: there has been earlier work [1] on designing differentiable algorithm for top-k operation. Different from this work, [1] views the top-k as the solution to an Optimal Transport problem. I am not an expert in this field, but could the authors discuss the difference between this work and [1].

* Q3: Actually, the experiment on the  large-scale, real-world recommender system looks more interesting to me. I wonder why the authors put them in the appendix (perhaps because of the page limit?).

* Q4: I notice that the submission history was "hidden as part of the resubmission bias experiment". Could the authors provide more information about that?

[1] Differentiable Top-k Operator with Optimal Transport, NeurIPS 20. https://arxiv.org/abs/2002.06504




**Time Spent Reviewing:**

5

---

> ### Author Response · Authors · 2021-08-10
> **Reply to Reviewer 1DHt**
>
> Thank you for taking the time to review the paper and for your feedback. Please find point-by-point responses to your questions below.
>
> Q1. While we did not perform experiments on language modeling tasks, we agree that Top-k might be faster during training if the same sparsity level is used for both Top-k and DSelect-k. However, based on the datasets we considered, DSelect-k may be able to achieve better predictive performance and lead to sparser models. We also note that the speed of convergence of DSelect-k (so that it can start performing conditional training like Top-k) can be controlled through the entropy regularization hyperparameter.
>
> Q2. Thank you for pointing this out. The approach in [1] presents a differentiable approximation of the Top-k operation. In our approach, we do not rely on a Top-k operation. We formulate the problem using a cardinality constraint and present an exact reparameterization that allows for transforming the constrained problem into an unconstrained one. We also note that the soft Top-k operation in [1] requires solving an optimization subproblem (using an iterative algorithm) in each call to the forward pass, whereas the forward pass in our approach can be computed in closed form. In the per-example gating setting, the forward pass in [1] will require solving a subproblem for each input example, which can be computationally prohibitive (especially for large batch sizes). We will cite [1] and add this discussion to the related work.
>
> Q3. Yes, we had very limited space. In the camera-ready version, we will move the real-world recommender system to the main paper, as we will get an additional page. Thank you for the suggestion.
>
> Q4. NeurIPS is performing an experiment in which the submission history is randomly hidden for some reviewers (we have no control over this).

---

### Official Review · Reviewer_XzFT · 2021-07-24

**Rating:** 6
**Confidence:** 4

**Summary:**

This work proposes Dselect-k, a continuously differentiable gate, to select a set of at most k out of n experts for the Mixture-of-experts (MoE) architecture. The Dselect-k based MOE can be trained end-to-end by the first-order optimization method, and it has promising performances on synthetic and different real-world MTL problems.


**Limitations And Societal Impact:**

Yes

**Main Review:**

I have reviewed this work before for ICML 2021, and now act as an opt-in reviewer with a very tight deadline. So my comments will mainly focus on the improvements/changes this work has made, as well as the unsolved concerns in my previous review.

Strengths:

+ This paper is well-written and easy to follow;

+ It is glad to see this work now has a clear discussion with the highly related stochastic subset selection methods. But there are still some concerns on the comparison (see below);

+ In addition to predictive performance, the obtained sparse gate can also lead to better model interpretability and computation savings on inference, which is desirable for real-world applications;

+ The proposed method has promising results on different MTL problems, especially the large-scale real-world recommender system application;

Concerns:

1. Comparison with Stochastic Subset Selection: The proposed Dselect-k is closely related to the stochastic subset selection approaches [1] for selecting a k-subset with gradient-based methods. As pointed out in the paper (and the appendix), there are some important differences that make Dselect-k a better choice for MTL-MOE applications considered in this work, such as the deterministic property and better interpretability. I appreciate the thorough discussion provided in the current manuscript, but still have some concerns on the comparison:

    - Experimental Comparison: This paper claims the deterministic/stochastic subset selection approaches are for different applications. However, as discussed in the appendix, there are closely related works that use the stochastic gumbel-softmax trick to select different components for an MTL neural network (as considered in this work) [2,3]. Indeed, this work had an experimental comparison with the method proposed in [3] in the earlier version. Why are those experimental results removed in this version? I still believe a careful experimental comparison with the stochastic subset selection is needed to truly show the pros/cons of the proposed method.

    - Analysis on Determinism and Interpretability: This work has a valid claim that the deterministic approach would be more suitable for applications that require better interpretability. It is understandable that making the simple Gumbel-softmax gate [3] deterministic at inference will lead to dense solutions or possibly all zeros, since the gumbel-softmax gate method has independent predictions for each gate. But is it still the case for the deterministic version of stochastic subset selection [1] where the output is always a k-hot vector?

2. Exact Control: It seems the proposed method can only guarantee selecting *at most* k items but not *precisely* k items as in Top-k and other k-subsect selection methods.

3. Good Performance v.s. Sparsity: It is still unclear why sparsity in gating is also good for task performance. I think the dense gate approach could provide a more robust prediction (kind of ensemble) at the cost of higher computation cost.

4. Experimental Results:
   - The large-scale real-world recommender system application is now moved to the appendix, and the remaining experiments (MovieLens and Multi-MNIST) are quite toy-ish. I believe the industry-level recommender system application will not be open source. It would be much better to compare the proposed method on the open-source and widely-used MTL problems, such as NYUv2[4], CityScapes[5], and Taskonomy [6].

   - In the MovieLens problem, the two tasks have different losses (classification and regression) in different scales. It is not suitable to only report the combined loss, especially when some of them are very close to each other (e.g., static DSelect-k and top-k). Why not report both the task 1/2 losses, or task 1 accuracy v.s. task 2 mse, as for MultiMNIST?

   - DSelect-k typically selects 1 ~ 2 experts in all experiments. When it selects one expert, it simply reduce to the shared bottom (all tasks share the same expert) or the single-task baseline (each task has its own expert). It would be good to have a single-task baseline for all experiments, which usually has better performance than many MTL baselines in the cost of one model (1 expert) for each task. See results in [7] for MultiMNIST, and MMOE (Ma et al., 2018) for UCI Census-income.


[1] Max B. Paulus., et al. Gradient Estimation with Stochastic Softmax Tricks. NeurIPS 2020.

[2] Krzysztof Maziarz., et al. Gumbel-matrix routing for flexible multi-task learning. arXiv preprint arXiv:1910.04915.

[3] Ximeng Sun., et al. Adashare: Learning what to share for efficient deep multi-task learning. NeurIPS 2020.

[4] Nathan Silberman., et al. Indoor segmentation and support inference from rgbd images. ECCV 2012.

[5] Marius Cordts., et al. The cityscapes dataset for semantic urban scene understanding. CVPR 2016

[6] Amir R Zamir., et al. Taskonomy: Disentangling task transfer learning. CVPR 2018.

[7] Ozan Sener and Vladlen Koltun. Multi-Task Learning as Multi-Objective Optimization. NeurIPS 2018.

**Time Spent Reviewing:**

4

---

> ### Author Response · Authors · 2021-08-10
> **Reply to Reviewer XzFT**
>
> Thank you for taking the time to review the paper and for your feedback. Below is a discussion on your concerns.
>
> 1. - In a previous draft of the paper, we included comparisons with the Gumbel-softmax-based approach of [2,3], and DSelect-k performed better overall. However, as discussed in the appendix, the approach of [2,3] cannot control the sparsity level explicitly and was not proposed for MoE models. In fact, the latter approach cannot perform per-example gating, which is a main topic in this paper. However, based on previous feedback, it seems that some readers thought that Gumbel-softmax solves the same problem as DSelect-k and Top-k. So to avoid such confusion and given the space constraints, we did not include comparisons with [2,3] in the current draft. We would be happy to include these results if the paper is accepted, as we will get one additional page. Also, we note that the major works on MoE consider softmax and Top-k gates, which we compare against.
>    - While we agree that the stochastic approach of [1] overcomes the all-zero issue, we note that it cannot be readily applied here for multiple reasons:
>      - It’s unclear what the choice of the noise distribution and the regularizer (f) should be. The authors of [1] caution against taking the choice of Gumbel noise by default. In some intended applications these choices are clear, but not in MoE.
>      - We cannot easily tune over f. Specifically, the approach in [1] requires solving a convex program (the SST) at each training step. Each choice of the regularizer (f) requires its own specialized solver for the convex program (e.g., squared L2 norm requires iterative gradient-based methods, functions from the exp. family require dynamic programming, ...).
>      - In the per-example gating setting, each example will require solving its own convex program (at each training step), which can be computationally prohibitive. In contrast, our approach allows for computing the forward/back passes in closed form, so it’s easy to implement and experiment with.
>
> 2. The problem we’re attempting to solve is cardinality constrained minimization (Problem 2 in the paper). We purposefully chose this formulation as opposed to selecting exact k because it can allow for sparser solutions. If the optimization algorithm does a good job, it might be able to find a model with < k experts with a (training) loss that is close to that of k experts---in such situations, the solution with < k experts may be preferable. The cardinality constraint we use  (<=k as opposed to exact k) has a long history in statistics, compressed sensing, and optimization for the reason mentioned earlier, among others. We will clarify this difference more in the related work section.
>
> 3. Thank you for raising this important point. We note sparse models are performing better in some cases, and not on all datasets. Our hypothesis is that dense MoE is using a relatively large number of parameters, which is leading to overfitting (even in settings where overparameterization is helpful, increasing the number of parameters too much can lead to overfitting). Sparsity can reduce the effective number of parameters to be estimated. Side note for intuition: In linear models, sparsity is theoretically known to mitigate overfitting in certain high-dimensional settings (e.g., it can lead to better prediction and estimation than dense models).
>
> 4. - We moved the real-world recommender system to the appendix because of space limitations. We will get this back to the main paper in the camera-ready version (as we will have one extra page).
>    - Thanks for the suggestion. We will add additional metrics to the MovieLens results.
>    - Our main goal in the experiments is to understand the differences in the gating performance between DSelect-k, Top-k, and softmax, for a fixed multi-task architecture. We do agree that there are many cases where single-task models will perform better (and these have been extensively documented in the literature as you pointed out).

---

> > ### Comment · Reviewer_XzFT · 2021-08-17
> > **Follow-up Questions**
> >
> > Thank you for your detailed and to-the-point response to my concerns, where many of them are properly addressed. I increase my score to 6, and have a few follow-up comments and questions:
> >
> > **1. Comparison with Stochastic Subset Selection:** Thank you for the detailed discussion, and I am now convinced that Dselect-K has its own advantage on the MTL-MOE (and per-example) setting considered in this work. I think the unique advantage of Dselect-K (e.g., closed-form forward/backward computation) should be explicitly discussed and emphasized in the paper.
> >
> > **3. Good Performance v.s. Sparsity:** I raised this concern because the contribution (iii) says: "... our gate outperforms state-of-the-art gates in terms of parameter sharing and *predictive performance*", where I believe the dense gate (Softmax MoE) is also one of the SOTA gates.
> >
> > For the predictive performance, I can get the point that sparsity (on the number of parameters) is good for a single overparameterized linear model with relatively small training samples (the large p, small n setting). However, I am not sure whether sparsity (on the number of models) is also good for the mixture/ensemble of multiple models with a sufficiently huge amount of data (huge n), which is the case for industry-level applications.
> >
> > Given the experimental results, is it suitable to say Dselect-K outperforms state-of-the-art gates in terms of predictive performance?
> >
> > **4. Experimental Results:**
> > - Along with the discussion on sparsity, it is interesting to know the performance of Top-k with k = 2 for Multi-MNIST/Multi-Fashion MNIST, which manually makes it have a similar number of experts with Dselect-k.
> > - What happens if we aggressively set K = 1 in Top-k and DSelect-k (kind of model selection in MOE)? I ask this one since Top-k outperforms Dselect-k in the (static, alpha = 0.1) case for MovieLens, which is the only case that Top-k has a smaller number of experts (in the rest case, DSelect has less than two experts).
> > - Experiment on Larger Dataset:
> >     - I asked for experiments on larger datasets (e.g., Taskonomy) because the per-example sparse-gated MoE is more suitable for the problem with sufficiently large training sets as discussed in the Top-k paper.
> >
> >     - The computational efficiency to train per-example gating is one crucial advantage of Dselect-k. However, the DSelect-k(per-example) only slightly outperforms DSelect-k(static) for MovieLens (2 out of 3 cases, marginal gain), and even performs worse in the Multi-MNIST/Multi-Fashion experiments.
> >
> >     - The only large-scale real-world recommendation experiment shows DSelect-k clearly outperforms Top-k on all eight different tasks, but not for per-example gating v.s. static gating. In addition, I believe this experiment will not be open source. Given the above discussion, I think the current experiment results are still relatively weak in this work.

---

> > > ### Author Response · Authors · 2021-08-19
> > > **Follow-up**
> > >
> > > Thank you for the discussion and for increasing the score.
> > >
> > > 1\. We will emphasize the advantage of the closed-form expressions for the forward/backward passes in the introduction and the related work section. Thank you again for raising this point; we think that including this discussion will strengthen the paper's message.
> > >
> > > 3\. We agree that softmax leads to SOTA performance in many settings and can outperform DSelect-k or other sparse gates (Multi-MNIST is one example in our experiments). We also agree that for sufficiently large n, sparsity may not be helpful in terms of predictive performance. Whether sparsity is helpful or not depends on a combination of factors (including the ratio of n and p, the correlation structure of the data matrix, choice of the optimization algorithm, among others). We will qualify the statement you referred to and say that DSelect-k can be competitive in terms of predictive performance.
> > >
> > > 4\. Experiments:
> > >   - We tuned k for Top-k in {2,4}. Based on validation, the choice k=2 is expected to perform worse (on Multi-MNIST).
> > >   - We did not try the choice k=1 since the model will be forced to be either shared-bottom (which we compare with) or single-task. We also note that the values k=2 and k=4 were used by Shazeer et al. 2017.
> > >   - Regarding static vs. per-example gating:  We saw some improvements from per-example gating on the MovieLens dataset; although the difference is small, it seems to be significant (based on standard error). However, we should point out that we do not expect per-example gating to generally improve predictive performance. In [14], the MoE was introduced as a method for better interpretability and faster training (specifically, their experiments show that the local model typically converges faster than global models, due to expert specialization). The recent works on sparse (per-example) MoE are also mainly motivated by computational and interpretability considerations. In terms of generalization performance (bias/variance tradeoff), there doesn't seem to be a clear advantage from per-example MoE compared to classical (global) models, e.g., see [13]. In the introduction, we will emphasize that the main motivation behind per-example gating is interpretability and faster computation.
> > >
> > > References:
> > >
> > > [13] Robert A Jacobs. Bias/variance analyses of mixtures-of-experts architectures. Neural computation, 9(2):369–383, 1997.
> > >
> > > [14] Robert A Jacobs, Michael I Jordan, Steven J Nowlan, and Geoffrey E Hinton. Adaptive mixtures of local experts. Neural computation, 3(1):79–87, 1991.

---

> > > > ### Comment · Reviewer_XzFT · 2021-09-02
> > > > **My Final Score**
> > > >
> > > > Thank you for the follow-up response, I keep my positive score (6) for this work.

---

### Decision · Program_Chairs · 2021-09-27

**Decision:**

Accept (Poster)

**Comment:**

All four reviewers agreed that the paper is very well written, easy to read, the idea is novel, and the experimental results are convincing and consistent. In the discussion, there was no disagreement in accepting the paper. As the authors promised, I would like to encourage the authors to include comparisons against Gumbel softmax based apporachs or some other stochastic subset selection methods. It would be great to prepare text and figures to highlight the shortcomings of these approaches (e.g. impossibility of per-example gating etc.).